# DASH: DAta-Efficient Learned Cost Models for Sparse Matrix Computations on Emerging Hardware Platforms

## Abstract

Sparse matrix computations are becoming increasingly significant in deep learning and graph analytics, driving the development of specialized hardware systems known as accelerators to meet the growing need for optimized performance. Optimizing these computations, however, presents significant challenges due to their sensitivity to variations in input sparsity patterns and code optimizations. While ML-based cost models and search techniques have shown promise in optimizing sparse matrix computations in general-purpose hardware like CPUs, these cost models require large datasets for effective training. Collecting such extensive datasets is particularly impractical for emerging hardware platforms that only have access to expensive simulators in the early design stages. To overcome this, we propose DASH, which trains learned cost models using low-cost data samples from widely accessible general-purpose hardware (such as CPUs), followed by few-shot fine-tuning to efficiently adapt to emerging hardware platforms. DASH introduces a novel approach that leverages the homogeneity of input features across different hardware platforms while effectively mitigating heterogeneity. This enables DASH to achieve comparable accuracy using only 5% of the data samples required by a cost model trained exclusively using data samples from an accelerator. We evaluate DASH on two critical sparse operations—SpMM and SDDMM—on an emerging sparse accelerator using 715 distinct sparsity patterns. Our experimental results show that DASH outperforms existing techniques that use transfer learning by 28.44%, achieving average speedups of $1.47\times$ (up to $5.46\times$) for SpMM and $1.39\times$ (up to $4.22\times$) for SDDMM.

## 1 Introduction

Sparse matrix computations have gained increased significance with the recent advancements in sparse deep learning and graph analytics (Beltagy et al. (2020); Ye & Ji (2021); Child et al. (2019); Dao et al. (2021)) workloads. As a result, many hand-crafted performance optimization techniques have been suggested in the literature that improve the runtime performance of sparse matrix computations (Kjolstad et al. (2017); Ye et al. (2023); Hong et al. (2019); Jiang et al. (2020)). These computations use compressed sparse formats (e.g., compressed sparse row) to only compute on non-zero values of matrices. Since the non-zero distribution of values in these matrices can vary, it has been challenging to come up with performance optimizations that work well across diverse inputs.

To overcome this challenge, machine learning (ML)-based program optimization techniques have been introduced to optimize sparse matrix computations on established hardware platforms such as CPUs and GPUs (Won et al. (2023); Yang et al. (2023)). These techniques adaptively select a program configuration based on the input sparse matrix features. For example, WACO (Won et al. (2023)) introduced learned cost models to predict the runtime cost of programs under different sparse matrices and program configurations. It then used search-based techniques to automatically find the optimal program configuration based on the cost model output. Overall, these ML-based techniques show superior performance and adaptability across a diverse range of inputs compared to manually crafted performance optimization techniques.

Recently, on the hardware front, new domain-specific machines specifically designed for sparse operations are emerging (Aananthakrishnan et al. (2023); Gerogiannis et al. (2023); Hegde et al. (2019); Li et al. (2023); Jin et al. (2024)). These machines, known as hardware accelerators, offer significant speedups over established hardware platforms. Similar to CPUs and GPUs, sparse accel-

erators offer different program configurations to accommodate diverse sparse inputs (Gerogiannis et al. (2023); Jin et al. (2024); Li et al. (2023); Muñoz-Martínez et al. (2023); Gerogiannis et al. (2024)) that should be configured by software performance optimizations. However, unlike CPUs and GPUs, most emerging accelerators that are at early design stages, only have access to expensive simulators. This poses a significant challenge for computer architects, as exhaustively evaluating all potential program configurations to determine the best during the early stages is not feasible. Hence, in the presence of any inefficiencies observed during early stage simulations, it has become difficult to diagnose whether the inefficiencies stem from suboptimal hardware design choices or program configurations. Further, when designing hardware accelerators, it would be ideal to avoid over-provisioning hardware resources (e.g., increasing cache size) if such inefficiencies can instead be mitigated via improved software strategies (e.g., adopting a different tiling strategy). Therefore, the need to automatically select the optimal program configuration during the design space exploration (DSE) phase of accelerator development is important, as it eliminates a dimension of complexity.

The learned cost models used in ML-based optimization techniques targeted at CPUs and GPUs are trained with relatively *large* supervised datasets. Those datasets consist of program configurations and sparse matrices as inputs, and runtimes as labels. Usually, these datasets consist of hundreds of thousands of such labeled data items Won et al. (2023). Unfortunately, collecting datasets of comparable size for emerging hardware accelerators that have only simulators – which is the case for most – is prohibitively costly. The time needed for a simulation to complete is many orders of magnitude longer than the real execution of the program on the actual chip. For example, it can take up to **two weeks to collect a single data point** using the simulator of the state-of-the-art SPADE sparse accelerator (Gerogiannis et al. (2023)). At the same time, the same program would take less than a second to execute on the real chip when it is finally fabricated. Collecting large datasets would require huge clusters running simulations for months or even years. Therefore, in order to bring the same benefits of ML-based optimizations to accelerator platforms at their early stages, we need to rethink learned cost model construction techniques that are data-frugal and highly sample-efficient.

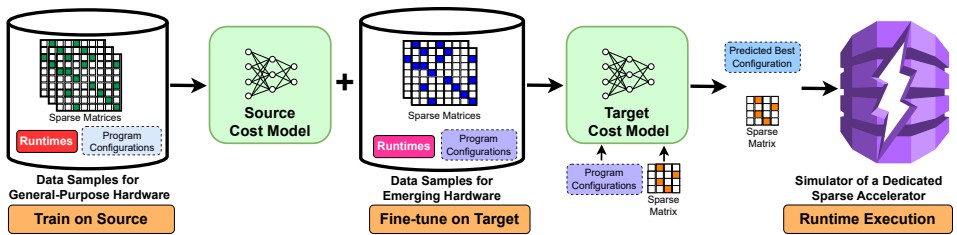

Figure 1: Transfer learning pipeline of DASH.

Inspired by the success of transfer learning in other domains (Weiss et al. (2016); Zhuang et al. (2020)), researchers have proposed different transfer learning techniques to reduce data requirements for training cost models (Sasaki et al. (2022); Zheng et al. (2021)). The proposed solution is to leverage knowledge transferred from cost models learned on one hardware platform (source) to another (target) using the ubiquitous pre-train and fine-tune paradigm (Krizhevsky et al. (2012)). Such techniques have shown to reduce the data requirement for the target platform. Therefore, potentially using such techniques to transfer cost models learned on general-purpose hardware platforms to emerging accelerator platforms can reduce the number of data points we need to collect from expensive simulations (Figure 1). However, we notice that most of the previous works have achieved effective knowledge transfer only between general-purpose hardware platforms of *the same type* (e.g. CPU-to-CPU, GPU-to-GPU)( Sasaki et al. (2022),Won et al. (2023), Zheng et al. (2021)). Transferring between hardware of different types (e.g. CPU-to-accelerator) poses unique challenges.

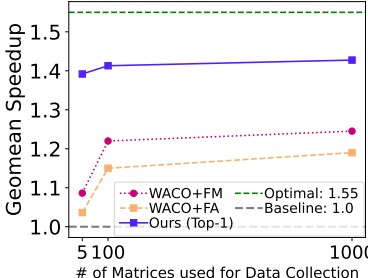

Figure 2: Geomean speedups from existing systems for SpMM on SPADE; WACO with Feature Augmentation (WACO+FA), and Feature Mapping (WACO+FM).

**Heterogeneous program configuration spaces.** The program configurations for emerging sparse accelerators – which become the input feature space of cost models – can be very different from those that are available for general-purpose hardware. For example, emerging sparse accelerator plat-

forms have software-managed buffers instead of hardware-managed caches and specialized, rather than general-purpose, pipelines. This disparity in program configuration spaces for general-purpose hardware and emerging accelerators makes it challenging to come up with highly accurate transfer learning techniques. Existing heterogeneous transfer learning techniques (Liang et al. (2019)), such as feature augmentation (Daumé III (2009); Duan et al. (2012)), offer a potential solution. However, these techniques often produce feature representations that are too sparse for the cost model to effectively learn, specifically when accommodating a diverse set of program configuration across different hardware platforms. Figure 2 shows the results of applying popular heterogeneous transfer learning techniques – feature augmentation (FA) and feature mapping (FM) – to a sparse learned cost model, WACO (Won et al. (2023)). Even when using data samples from 1000 matrices for fine-tuning on the SPADE accelerator, the best configurations found under WACO+FA and WACO+FM are far from the optimal. Therefore, we need better techniques to handle the heterogeneity of program configurations present across different hardware platforms.

**High sample efficiency requirement.** Existing transfer learning solutions for learned cost models that operate within homogeneous feature spaces, typically require at least 25% of the original dataset that was used in a non-transfer learning setup to achieve competitive performance on the target hardware platform (Sasaki et al. (2022)). The target dataset requirement for these solutions can further increase due to the heterogeneous input feature spaces between general-purpose hardware and emerging accelerators. This makes it infeasible to adopt existing solutions in their current form for accelerators in early design stages. Therefore, we need data-frugal techniques.

**DASH.** In this paper, to overcome these challenges we present DASH, a novel framework for developing learned cost models which enables effective knowledge transfer (Figure 1). DASH uses WACO's (Won et al. (2023)) neural cost model architecture as the base model (*WacoNet*) but incorporates key changes to make it amenable for transfer learning. DASH enables the discovery of better program configurations that are closer to the optimal (Figure 2), while requiring significantly less accelerator data samples for fine-tuning.

DASH is centered around two key principles introduced in Neyshabur et al. (2020): feature reuse and the capture of low-level statistical information. We observe that, even though the program configurations between general-purpose hardware and accelerators are heterogeneous, there are certain feature spaces that can be mapped due to their similarities. Motivated by this observation, we propose an **approximate mapping of comparable code optimizations**, effectively segregating the feature space generated by program configuration into homogeneous and heterogeneous components. This allows feature reuse across the source and target platforms. The heterogeneous components represent non-mappable hardware specific parameters that can be disparate across different platforms. Such components can introduce challenges during transfer learning due to negative transfer. To separately encode the heterogeneous feature spaces, we introduce a **novel latent space representation of the heterogeneous input feature space** using an auto-encoder. This novel formulation of the feature space allows us to effectively reuse features while minimizing the impact of negative transfer. Further, we observe that certain layers of *WacoNet* do not contribute heavily to the final prediction and this over-parameterization can hinder transferability due to over-fitting. To mitigate this, DASH modifies *WacoNet* by reducing the number of layers and extracting features at various depths and scales, effectively allowing the model to capture low-level statistical information.

We evaluate DASH on two widely used sparse operations, Sparse Matrix-Matrix Multiplication (**SpMM**) and Sampled Dense-Dense Matrix Multiplication (**SDDMM**), by transferring from a CPU to an emerging sparse accelerator SPADE (Gerogiannis et al. (2023)). Our experimental results show that DASH outperforms existing techniques that leverage transfer learning by 28.44%, achieving an average speedup of 1.47× (up to 5.46×) for SpMM and 1.39× (up to 4.22×) for SDDMM. To further demonstrate the generalizability of our approach, we transferred from a CPU to a GPU, achieving an average speedup 1.17× (up to 1.61×) for SpMM and 1.15× (up to 1.49×) for SDDMM.

In summary, this paper makes the following contributions.

- We introduce techniques to segregate and encode the homogeneous (approximate mapping of comparable code optimizations) and heterogeneous (latent representation using an auto-encoder) components of program configurations across different hardware platforms used for sparse matrix computations.
- We introduce DASH, a novel framework for developing learned cost models that are amenable to few-shot fine-tuning across different hardware platforms, leveraging above techniques.

- We evaluate and show that DASH produces highly accurate transfer learned cost models for emerging sparse accelerators at early design stages with minimal data collection overhead. Specifically, we demonstrate that DASH achieves average speedups of $1.47\times$ on SpMM and $1.39\times$ on SDDMM on the state-of-the-art sparse accelerator SPADE. Furthermore, we perform additional experiments and ablation studies to demonstrate its benefits and generalizability.

## 2 BACKGROUND AND RELATED WORK

### 2.1 SPARSE MATRIX COMPUTATIONS

Sparse matrix computations perform computational tasks that involve tensors where most of the elements are zero. These computations are optimized to efficiently process only the non-zero values. We describe two operations frequently used in these computations below.

**Sparse Matrix-Matrix Multiplication (SpMM)** is the operation of multiplying a sparse matrix $\mathbf{A} \in \mathbb{R}^{M \times K}$ with a dense matrix $\mathbf{B} \in \mathbb{R}^{K \times N}$, resulting in an output matrix $\mathbf{D} \in \mathbb{R}^{M \times N}$. The SpMM operation can be expressed as $D_{i,k} = \sum_j A_{i,j} \cdot B_{j,k}$, where $A_{i,j} \neq 0$.

**Sampled Dense-Dense Matrix Multiplication (SDDMM)** is an operation that involves the multiplication of two dense matrices, followed by an elementwise multiplication with a sparse matrix. Given a sparse matrix $\mathbf{A} \in \mathbb{R}^{M \times N}$, a sparse output matrix $\mathbf{D} \in \mathbb{R}^{M \times N}$, and two dense matrices $\mathbf{B} \in \mathbb{R}^{M \times K}$ and $\mathbf{C} \in \mathbb{R}^{K \times N}$, SDDMM operation can be expressed as $D_{i,k} = A_{i,k} \cdot \sum_j (B_{i,j} \cdot C_{j,k})$, where $A_{i,k} \neq 0$.

### 2.2 SPARSE MATRIX PROGRAMMING SYSTEMS AND HARDWARE

Sparse matrix computations can be executed on a variety of hardware platforms, including CPUs, GPUs, and dedicated sparse accelerators. The execution strategy for these computations depends on both the hardware platform and the corresponding programming system used. In this work, for CPU execution, we use TACO (Kjolstad et al. (2017)), a domain-specific language and a compiler designed for sparse tensor algebra and optimized for CPU. For GPU execution, we employ SparseTIR (Ye et al. (2023)), a sparse tensor compilation framework developed as an enhancement to TVM's Tensor IR(Chen et al. (2018a).) As our dedicated sparse accelerator, we use SPADE (Gerogiannis et al. (2023)), which has a tiled-based instruction set architecture (ISA) to leverage different variations of SpMM and SDDMM operations. Throughout the remainder of this paper, we will refer to this dedicated sparse accelerator as SPADE.

Table 1: Configurable program configuration parameters available across CPU, GPU, and SPADE.

| Configurable Parameters | CPU | GPU | SPADE | Type |
|---|:---:|:---:|:---:|:---:|
| Loop Strip-mining | ✓ | ✓ | | Numerical |
| Loop Reordering | ✓ | ✓ | | Categorical |
| Format Reordering | ✓ | | | Categorical |
| Loop Binding | | ✓ | | Categorical |
| Loop Unrolling | | ✓ | | Categorical |
| Tiling | | | ✓ | Numerical |
| Barrier | | | ✓ | Binary |
| Cache Bypassing | | | ✓ | Binary |
| Matrix Reordering | | | ✓ | Binary |

A sparse matrix programming system supports a range of code optimizations that modify the structure of the code to enhance performance. The effectiveness of these code optimizations depends on assigning specific values to the parameters of the program configuration. By tuning these parameters, we can significantly impact the runtime performance of sparse operations. Table 1 outlines the configurable parameters available in program configurations for different hardware platforms explored in this work. Further details on related code optimizations can be found in Appendix B.

### 2.3 ML-BASED COST MODELS

**Learned Cost Models.** Cost models act as fast and cost-effective proxies for executing workloads on real hardware. Their primary goal is to accurately estimate the execution time of workloads as they would perform on real hardware. To achieve this, these cost models can be trained on data samples with various program configurations and then used to predict the program configuration

that will deliver the optimal performance. Hence, generally, the training objective of cost models is tied with minimizing $|t^*_{CM} - t^*|$, where $t^*$ is the runtime of the true optimal program configuration and $t^*_{CM}$ is the runtime of the best program configuration suggested by the cost model (***accuracy objective***)(detailed Appendix A). Finding the best configuration suggested by the cost model is usually done using auxiliary intelligent search techniques such as simulated annealing, Monte Carlo tree search, and reinforcement learning.

There have been numerous works on learned cost models to predict the runtime of different workloads targeting different hardware platforms Chen et al. (2018b); Adams et al. (2019). These techniques range from simple XGBoost (Chen & Guestrin (2016)) based cost models Chen et al. (2018a) to sophisticated deep neural network based models (Baghdadi et al. (2021); Kaufman et al. (2021); Zhai et al. (2023); Zheng et al. (2020); Won et al. (2023)). WACO Won et al. (2023) is a learned cost model specifically built for sparse matrix computations that we use as our base cost model.

**Transfer Learning.** Transfer learning is a technique that leverages knowledge gained from a task in a source domain to improve the performance of a related task in a target domain, where data collection can often be challenging (Bozinovski (2020)). There have been many successful examples of transfer learning techniques in a wide range of fields (Weiss et al. (2016)). Transfer learning can be categorized into two main types: homogeneous transfer learning (Zhuang et al. (2020)), where the input and label spaces are the same, and heterogeneous transfer learning (Day & Khoshgoftaar (2017)), where either one or both can be different. In program optimization, transfer learning has been successfully used to transfer cost models learned from one hardware platform to another, primarily in homogeneous settings, to minimize the target domain data requirements Zheng et al. (2021); Sasaki et al. (2022). In this work, we seek to minimize the target domain data requirement during fine-tuning (Shen et al. (2021)), by targeting heterogeneous input feature spaces present between general-purpose hardware and emerging sparse accelerators (***data-collection objective***))(detailed in Appendix A).

# 3 OUR METHODOLOGY: DASH

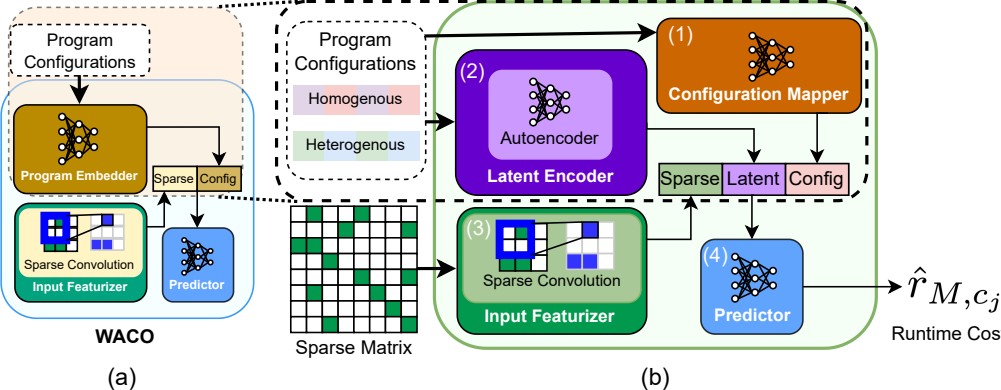

Figure 3: A comparative overview of the enhanced cost model design in DASH (b) alongside WACO's cost model design (a), highlighting key differences.

In this section, we present DASH, a novel framework to design data-efficient learned cost models to accelerate the execution of sparse matrix computations on emerging hardware platforms. The following subsections outline our contributions toward achieving the objectives set forth in Section 2.3; *maximizing the accuracy while minimizing the data collection overhead*. In Section 3.1, we review WACO's cost model, which serves as the base model for our work. Section 3.2 provides an overview of our key innovations, highlighting the enhancements in the cost model design that enable transfer learning. In Section 3.3, we explore how the homogeneity of program configurations is leveraged through the traits of comparable code optimizations. Finally, Section 3.4 addresses how we handle the heterogeneity in program configurations introduced by hardware-specific code optimizations.

## 3.1 WACO'S COST MODEL

WACO (Figure 3(a)) (Won et al. (2023)) introduced the concept of directly feeding sparsity patterns as raw data into cost models. WACO's cost model employs submanifold sparse convolution net-

works (SCNN) (Graham & Van der Maaten (2017)) to extract features using the *input featurizer*. It leverages a neural network-based *program embedder* to capture the impact of code optimizations on sparse operations by encoding program configurations into embeddings. The program embeddings are merged with the extracted sparsity pattern features produced by the input featurizer. The merged inputs are then processed through a multi-layer perceptron *predictor* to estimate the execution cost.

## 3.2 Overview of Enhancements to Enable Transfer Learning

We build upon WACO considering it as our base model by refining its architecture to better facilitate transfer learning across diverse hardware platforms. Our improved cost model design (Figure 3(b)) is structured around four key components: *configuration mapper*, *input featurizer*, *latent encoder*, and *predictor*. The *configuration mapper* (Figure 3(b)(1)) and *latent encoder* (Figure 3(b)(2)) replace the *program embedder* in WACO, while the *input featurizer* (Figure 3(b)(3)) has been modified to more effectively capture low-level information from sparsity patterns. Both the *configuration mapper* and the *input featurizer* remain consistent across hardware platforms, serving as key components that enable efficient knowledge transfer.

**Configuration Mapper** ($\mathcal{FM}$). The configuration mapper captures homogeneity across hardware platforms by processing program configurations ($c_j$) and their parameters to identify similarities in code optimizations across various platforms. We designed it to approximately map similar configuration parameters across different hardware platforms (described in Section 3.3) to a unified feature space. This is achieved by using explicit mapping functions. The resulting parameters are subsequently passed through a multi-layer perceptron (MLP) to produce the final *configuration vector* $p_j$. In this work, we approximate the code optimizations *loop strip-mining* and *loop reordering* as $p_j = \mathcal{FM}(\phi(\cdot), \pi(\cdot), c_j)$. using the mapping functions $\phi$ and $\pi$, as detailed in Section 3.3.

**Input Featurizer** ($\mathcal{IFE}$). Matrices with identical dimensions and non-zero elements can exhibit vastly different sparsity patterns, making it difficult to extract meaningful features based only on statistical properties. Building on WACO's *input featurizer* Won et al. (2023), we modify the network architecture (Figure 3) to more effectively capture low-level information from sparsity patterns. Our network consists of 12 SCNN layers (compared to 14 layers in WACO), arranged in 4 blocks, each containing 3 sparse convolution layers. At the end of each block, we apply max pooling to condense spatial information. We increase the number of channels across blocks up to 256, whereas WACO kept them fixed at 32. These additional channels enables our design to capture hierarchical features more effectively throughout the network compared to WACO. For a given sparse matrix $M$, our *input featurizer* generates a *sparse feature vector* $s_M$, expressed as $s_M = \mathcal{IFE}(M)$.

**Latent Encoder** ($\mathcal{LE}$). We handle the heterogeneity of program configurations across hardware platforms using per-target autoencoders that compress the heterogeneous components of the configurations into compact latent representations (described in Section 3.4). An autoencoder is trained for each target–sparse primitive pair. During both training and inference, the *latent encoder* $\mathcal{LE}$ processes a configuration ($c_j$), transforming it into a latent representation $z_j = \mathcal{LE}(c_j)$, that encapsulates the unique characteristics of the program configuration.

**Predictor** ($\mathcal{P}$). As the final component of the cost model, the *predictor* (Figure 3(b)(4)) integrates the three feature vectors from the preceding components into a single unified vector, encapsulating all key information about the sparsity pattern and program configuration. This unified vector $(s_M \| p_j \| z_j)$ is passed through a multi-layer perceptron (MLP) to eventually output a scalar value representing the predicted execution cost, which can be expressed as $\hat{r}_{M,c_j} = \mathcal{P}(p_j \| s_M \| z_j)$.

## 3.3 Exploiting Homogeneity: Approximate Mapping of Code Optimizations

Different hardware platforms often use distinct programming systems, leading to variations in how code optimizations are parameterized (Figure 1). Further, an optimization available in one platform may not be directly available on another, requiring the combination of multiple other code optimizations to replicate the same impact. For example, *loop strip-mining* code optimization on CPUs can be closely approximated by collectively applying *barrier* and *tiling* optimizations in SPADE. By mapping the effects of these code optimizations using their program configuration parameters, we can expose patterns that facilitate effective knowledge transfer across hardware platforms. In this section, we present our approaches for approximately merging *loop strip-mining*, *barrier*, and *tiling* optimizations between CPU and SPADE, and *loop reordering* optimization between CPU and GPU.

***Loop strip-mining*** is a code optimization that decomposes large software loops into smaller segments to optimize computations for memory utilization and cache performance. In our context, it is applied to loops iterating over the indices $i$, $j$, and $k$ of matrices in SpMM and SDDMM sparse operations (Section 2.1), where parameters $I$, $J$, and $K$ are used to split these loops into outer and inner segments, resulting in a loop nest of six decomposed loops. The resulting loop segments are $\{i_1, i_2, j_1, j_2, k_1, k_2\}$ and their execution order is denoted by $\omega$. In SPADE, we approximate this using *barrier* and *tiling* optimizations. *Tiling* decomposes a matrix into smaller blocks to optimize data reuse in the local memory, while *barrier* controls the execution order of tiles. For example, enabling barrier optimization pauses the tiles scheduled by a control processing element until all previous tiles have been completed (Gerogiannis et al. (2023)). Similar to strip-mining parameters, the tiling parameters for $i$, $j$, and $k$ indices of matrices are represented in SPADE as $p_{\text{col}}, p_{\text{row}}, d_{\text{split}}$, respectively, while *barrier* is represented by $b$, where $b = 1$ if barrier is enabled, and $b = 0$ otherwise. Intuitively, tiling divides computations into smaller blocks, while barriers control synchronization during execution. By enabling and disabling barriers for various tiling configurations, we can dictate the order of computation. This resembles loop strip-mining and reordering in CPUs, where optimizing loop execution enhances performance and cache utilization. We can approximately map tiling and barrier parameters to the corresponding strip-mining parameters using the mapping function $\phi : \{p_{\text{col}}, p_{\text{row}}, s_{\text{split}}, b\} \rightarrow \{I, J, K, \omega\}$ as follows:

$$\phi(p_{\text{col}}, p_{\text{row}}, s_{\text{split}}, b) = (I, J, K, \omega)$$

$$I \approx p_{\text{col}}, \ J \approx p_{\text{row}}, \ K \approx s_{\text{split}}; \ \omega = \begin{cases} [k_2, \ j_2, \ i_2, \ i_1, \ j_1, \ k_1] & \text{if } b = 1 \\ [k_2, \ i_2, \ j_2, \ i_1, \ j_1, \ k_1] & \text{if } b = 0 \end{cases}$$

***Loop reordering*** is a code optimization that adjusts the execution order $\omega$ of loops to improve cache efficiency and facilitate parallel processing. It is often applied after loops are strip-mined. Here, we examine how this can be approximated for both CPU ($a_1$) and GPU ($a_3$). In CPU, loop strip-mining results in six decomposed loops $\{i_1, i_2, j_1, j_2, k_1, k_2\}$. Similarly, in GPU, loop strip-mining produces six loop segments, but the loop structure differs $\{i_1, i_2, j, k_1, k_2, k_3\}$ due to architectural changes of the platform. We approximate them using $\Omega(\cdot)$ function that determines the index of a loop segment and a mapping function $\pi_{a_i} : \{i_1, i_2, \ldots, k_2, \omega_{a_i}\} \rightarrow \{i_1, i_2, \ldots, k_3, \omega'_{a_i}\}$ as follows:

$$\pi_{a_1}(i_1, i_2, j_1, j_2, k_1, k_2, \omega_{a_1}) = (i_1, i_2, j_1, j_2, k_1, k_2, k_3, \omega'_{a_1}); \quad k_3 = 1, \quad \Omega_{a_1}(k_2) + 1 = \Omega_{a_1}(k_3)$$

$$\pi_{a_2}(i_1, i_2, j, k_1, k_2, k_3, \omega_{a_3}) = \left(i_1, i_2, j, j', k_1, k_2, k_3, \omega'_{a_3}\right); \quad j' = 1, \quad \Omega_{a_3}(j) + 1 = \Omega_{a_3}(j')$$

## 3.4 Mitigating Heterogeneity: Encode Hardware-specific Code Optimizations

While we can use the strategies described in Section 3.3 to approximate code optimizations with homogeneity, such techniques are not applicable to hardware-specific code optimizations. An existing approach for representing hardware-specific code optimizations across different hardware platforms is to encode them using feature augmentation. However, this results in excessively sparse feature vectors, as code optimizations that are not applicable to a selected hardware platform are zeroed out. Training models on such sparse feature vectors often leads to sub-optimal performance (Figure 15).

To address this limitation, we propose indexing the parameters of the heterogeneous component of the program configurations for each platform $a_i$ using low-dimensional latent representations. Specifically, we train an autoencoder $\mathcal{AE}_{a_i}$ to learn a latent representation $z_j$ for each configuration $c_j \in C_{a_i}$. This is accomplished by determining the value ranges for all parameters of the heterogeneous component in the program configurations, followed by training an autoencoder to learn an unsupervised embedding of this parameterization. Once trained, we use the encoder $\mathcal{LE}_{a_i}$ in $\mathcal{AE}_{a_i}$, which takes each configuration ($c_j$) as input and transforms it into its corresponding latent representation $z_j$, where $z_j$ is a fixed-size vector. By compressing configurations from different hardware platforms—each with varying parameters and ranges—into fixed-size vectors, we standardize the input for hardware-specific optimizations into the cost model. This compression significantly simplifies the model compared to feature augmentation, as the cost model now processes fewer input features, reducing its computational complexity. Since the hardware-specific optimizations from different hardware platforms are now represented in a unified latent feature space, it becomes feasible to capture any similarities in how they impact performance, which can then be leveraged in fine-tuning. Finally, this approach facilitates the seamless integration of emerging hardware platforms into DASH, as we can extend DASH to support new target hardware platforms by training new autoencoders and fine-tuning, eliminating the need to retrain the source model from scratch.

# 4 EVALUATION

## 4.1 DATASET, TRAINING AND EVALUATION SETUP

**Dataset.** Our experiments were conducted using real-world sparse matrices sourced from the SuiteSparse Matrix Collection (Davis & Hu (2011)). This dataset has been widely used in previous work (Hong et al. (2019); Jiang et al. (2020); Won et al. (2023)) and covers a broad spectrum of domains, ensuring a realistic and comprehensive evaluation of DASH's performance. To collect the training dataset, we performed the sparse matrix operations SpMM and SDDMM on three distinct hardware platforms: an Intel Xeon Gold 6348 **CPU** with 1TB of RAM, an NVIDIA A100 **GPU** paired with an Intel Xeon Platinum 8358, and **SPADE**, a simulated sparse accelerator with 32 processing elements operating at 0.8GHz.We gathered data samples using 1500 matrices for each hardware platform to use for model training and validation. For each matrix, we randomly sampled 100 program configurations to have diverse and representative training datasets across all platforms.

**Baselines and Implementation.** We executed SpMM and SDDMM on CPU, GPU, and SPADE using the respective programming systems introduced in Section 2.2. We used the default optimizations of these programming systems as our baseline environment. We implemented DASH in PyTorch, utilizing MinkowskiEngine (Choy et al. (2019)) to handle sparse convolution. Separate models were developed for SpMM and SDDMM to conduct precise performance predictions.

**Cost Model Evaluation.** We evaluated DASH's performance on 715 real-world matrices from the SuiteSparse Matrix Collection, ensuring that none of the evaluation data samples overlapped with the training set. For each matrix, we predicted the runtime cost across all program configurations and selected the top-1 and top-5. For each of the top-1 and top-5, we executed the selected program configurations on the target platform and identified the one with the shortest runtime. We then compared our results to the normalized runtime of the baseline executions, *WacoNet* with feature augmentation, and *WacoNet* with feature mapping by calculating the geometric mean (geomean) speedups for each to quantify DASH's overall effectiveness.

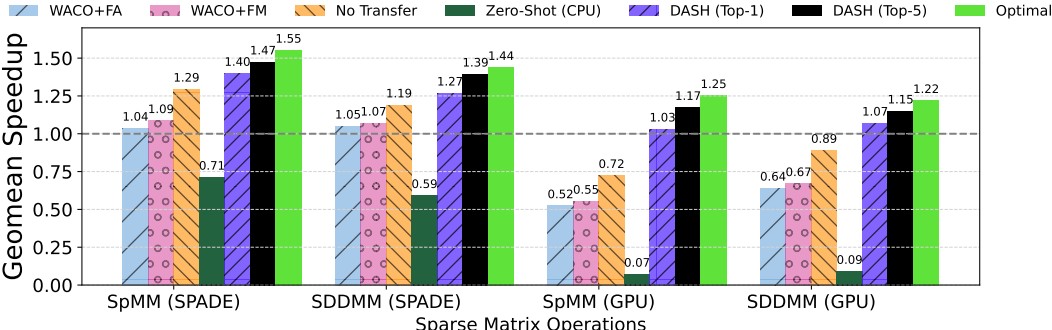

Figure 4: Geomean speedups of DASH and other techniques, normalized to the baseline.

## 4.2 TRANSFERABILITY OF COST MODELS

**Pre-training and Fine-tuning Procedure.** We trained source models on CPU using data samples from 100 matrices. The matrices were randomly selected from the training set while ensuring a balanced representation of their dimensions and sparsity. We empirically demonstrate in Section 4.3 (Figure 11) that training the source model with 100 matrices struck an optimal balance. Once the source model was trained, we performed few-shot fine-tuning on SPADE with only using data samples from 5 matrices. This decision was made to achieve the best trade-off between our objectives for accuracy and data collection (detailed in Section 4.3 (Figure 12)). We chose to train the source model on CPU since *WacoNet* was originally trained on it and it offers widespread accessibility.

**Transferability to SPADE.** Figure 15 illustrates the geomean speedups achieved using multiple techniques: zero-shot inference from the source model (zero-shot), a model trained exclusively on the target hardware using the fine-tuning dataset (no transfer), *WacoNet* with feature augmentation (WACO+FA), *WacoNet* with feature mapping (WACO+FM), and DASH's performance for both the top-1 and top-5 (k-best) predicted program configurations. Our results show that DASH consistently outperformed other techniques across both sparse operations and hardware platforms. Specifically for SPADE, DASH (Top-1) achieved an average speedup of 1.40× for SpMM, reaching 90% of

the optimal speedup of 1.55×. When expanding DASH (Top-5), it delivered an average speedup of 1.47×, achieving 95% of the optimal speedup. The optimal speedup was determined by running all possible configurations for the evaluation matrices and selecting the fastest execution time for each matrix. Similarly, for SDDMM in SPADE, DASH (Top-1) achieved an average speedup of 1.27× and DASH (Top-5) achieved an average speedup of 1.39×. This emphasizes DASH's ability to consistently find near-optimal program configurations with minimal fine-tuning across multiple sparse operations. The speedup gained for zero-shot inference from the source model was significantly lower than the baseline. In contrast, fine-tuning on a few data samples from SPADE led to significant performance gains demonstrating DASH's effectiveness in transferring knowledge.

**Transferability to GPU.** To further showcase DASH's ability to transfer knowledge across different hardware platforms, we extended our evaluation to GPU (Figure 15). The speedup trends on GPU aligned with those observed on SPADE, reinforcing the effectiveness of DASH. DASH (Top-1) delivered an average speedup of 1.03× and DASH (Top-5) yielded an average speedup of 1.17× for SpMM, with the optimal achievable speedup being 1.25×. In comparison, cusparseSpMM achieved a lower average speedup of 1.01×. For SDDMM, DASH (Top-1) resulted in an average speedup of 1.07×, while DASH (Top-1) yielded a 1.15× speedup, with the optimal being 1.22×. Zero-shot inference on the GPU was significantly worse compared to Zero-shot for SPADE, with speedups falling well below the baseline. This discrepancy is likely due to the larger inherent architectural differences between the CPU and GPU. Further, we evaluated the end-to-end performance impact on GPU for graph convolutional networks. Our results had a 1.06× overall speedup over DGL when training for 100 epochs with DASH (Top-1) on the Wiki-Talk matrix (∼2M rows) from the test set.

**Comparison with Other Transfer Learning Techniques.** For comparisons, we modified *WacoNet* to support feature augmentation and feature mapping, as it is not inherently optimized for heterogeneous transfer. Despite these modifications, DASH consistently outperformed both. For SpMM on SPADE, WACO+FA had an average speedup of 1.04×, while WACO+FM resulted in a slightly higher average speedup of 1.09×. In comparison, DASH delivered an average speedup of 1.40×, outperforming its closest alternative (WACO+FM) by 28.44%. The sub-optimal performance of WACO+FA and WACO+FM can be attributed to the increased sparsity in the feature space due to feature augmentation and their limited capacity to effectively mitigate the heterogeneity.

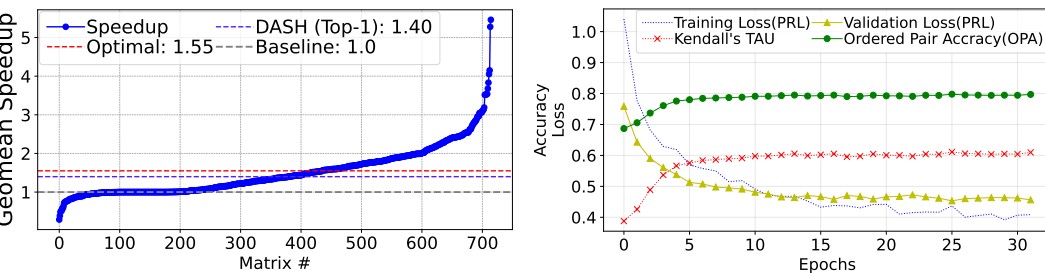

Figure 5: DASH per-matrix speedups (SpMM).    Figure 6: Loss and accuracy during training.

## 4.3 ADDITIONAL EXPERIMENTS AND ABLATION STUDIES FOR SPMM ON SPADE

**Speedup Performance.** Figure 5 shows the speedups achieved by DASH (Top-1) across all evaluated matrices. Matrices where the baseline outperformed DASH are indicated below the y = 1 dotted line. While the baseline outperformed DASH on a few matrices, the overall results demonstrate that DASH delivered substantial speedups (as high as 5.46x) for the majority of the dataset.

**Cost Model Accuracy.** Figure 6 shows the accuracy of DASH's cost model across training epochs using Pairwise Ranking Loss (PRL), Ordered Pair Accuracy (OPA), and Kendall's Tau (K-Tau). The steady decline in PRL for both training and validation loss indicates that the model effectively learns to rank program configurations without over-fitting. OPA and K-Tau demonstrated steady improvement, reaching 0.80 and 0.61, respectively, indicating that the training process is effective.

**Component-Level Contributions.** The effectiveness of our cost model relies on the inclusion of all components, each contributing uniquely to the overall performance. As illustrated in Figure 7, the exclusion of individual components leads to a noticeable decline in speedups. For example, excluding the *input featurizer* ($\mathcal{IFE}$) causes a decline from 1.40x to 1.26x. Similarly, omitting the *configuration mapper* ($\mathcal{FM}$) leads to a further decline to 1.16x, and excluding *latent encoder* ($\mathcal{LE}$)

lowers speedup to 1.01x. This emphasizes that each component contributes uniquely to the model's high performance, and all need to act synergistically to maximize the benefits of knowledge transfer.

**Selection of MLP Predictor.** As shown in Figure 3, the MLP predictor from WACO's base cost model was retained in our enhanced design. Figure 8 provides a comparative analysis of alternative predictors, including LSTM, GRU, and Transformer (TF). The results demonstrate that our proposed cost model design outperforms the alternatives, with the TF predictor achieving the next best performance with 1.36× speedup. These findings highlight that an MLP predictor is sufficient to deliver robust performance with limited data. In contrast, the suboptimal performance of the TF predictor can be attributed to the limited dataset size, as the high simulation costs associated with emerging hardware make it challenging to collect larger datasets for fine-tuning.

**Selection of Auto-Encoders.** Figure 9 shows our investigation into various methods for handling the heterogeneous components of program configurations. We evaluated choices ranging from conventional feature augmentation (FA) to principal component analysis (PCA), auto-encoders, and variational auto-encoders (VAE). Our findings reveal that auto-encoders were the most effective for capturing heterogeneous optimizations in a latent space. This was further supported by the smaller validation loss observed when training the auto-encoder to learn these latent representations.

**Data Collection Overhead w/o Transfer Learning.** Figure 10 shows that without transfer learning, the overhead of data collection becomes significant on emerging hardware due to the high costs of running simulations. For example, models trained exclusively on SPADE would require 20×–200× more target data samples (collected using 100–1000 matrices) to match or surpass the speedups achieved through DASH via transfer learning.

**Impact of Negative Transfer.** Figure 11 shows that using a large dataset to train the source model (e.g., data samples from 1000 matrices) does not necessarily lead to better outcomes. As the size of the training dataset increases, the model becomes overly specialized to the source platform, diminishing its adaptability during fine-tuning. For example, training on the CPU (source) with data from 100 matrices and fine-tuning on SPADE (target) with data from 5 matrices produces the best results. In contrast, training the source model with data from 1,000 matrices yields sub-optimal performance. This highlights the importance of balancing the source model's training dataset to avoid over-specialization and minimize the impact of negative transfer.

**Number of Samples in Fine-Tuning.** In Figure 12, we show DASH's performance as fine-tuning data samples increase. Despite fine-tuning on data from 1,000 matrices, the maximum speedup saturates at 1.42×. We can achieve a comparable speedup of 1.40× with data from 5 matrices, which shows the diminishing returns associated with larger datasets. Further, the non-transfer learning setup achieved a marginally higher speedup of 1.43× when using data from 1,000 matrices. However, considering the significant data collection overhead, these marginal improvements are not deemed beneficial.

## 5 CONCLUSION

In this paper, we introduced DASH, a novel framework to develop data-efficient learned cost models to optimize sparse matrix computations for emerging hardware platforms. DASH leverages a unique technique that capitalizes on the homogeneity of input features across different platforms while effectively mitigating heterogeneity. This enables DASH to train cost models using low-cost data samples from widely accessible general-purpose hardware (such as CPUs) and then fine-tune them for emerging hardware platforms with few-shot learning. Our results demonstrate that DASH is able to achieve near-optimal accuracy while maintaining significant sample efficiency.

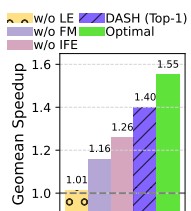

Figure 7: Ablation study for SpMM.

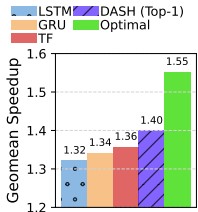

Figure 8: Selection of MLP predictor.

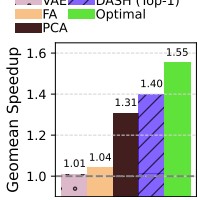

Figure 9: Selection of auto-encoders.

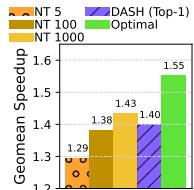

Figure 10: No transfer learning.

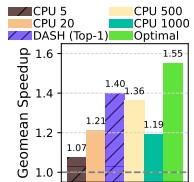

Figure 11: Impact of negative transfer.

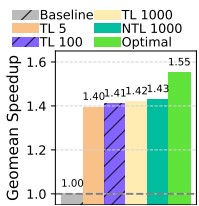

Figure 12: Impact of number of samples in fine-tuning.

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

# A PROBLEM FORMULATION

In this work, our aim is to build accurate learned cost models for emerging hardware platforms to enable faster identification of optimal program configurations. A key challenge is the need to maximize the accuracy of the cost model *(accuracy objective)* while using as few expensive (i.e. collected through simulation) data samples as possible *(data collection objective)*. We first formalize the program optimization objective and then tie it with the cost model objectives.

## A.1 PROGRAM OPTIMIZATION SELECTION

The goal of program optimization in sparse matrix computations is to select the optimal program configuration for a given hardware platform and input sparsity pattern from the total configuration space. Let configuration space $C_a$ be the set $\{c_1, c_2, \ldots, c_{m_a}\}$ of all valid program configurations for a given hardware platform $a$ ($m_a \in \mathbb{Z}^+$). For example, for CPU, a valid configuration from $C_{CPU}$ is a tuple of program configuration parameters for loop strip-mining, loop reordering, and format reordering (Table 1). The optimal program configuration minimizes the execution time of a sparse matrix computation. For an input sparse matrix (sparsity pattern) $M$, the optimal program configuration on platform $a$ can be given as, $c^* = \arg\min_{c_i \in C_a} \mathcal{T}_a(M, c_i)$, where $\mathcal{T}_a$ is the execution time function for platform $a$ (ground truth runtime). The execution time for the optimal program configuration is given by $t^* = \mathcal{T}_a(M_l, c^*)$.

## A.2 COST MODEL PERFORMANCE AND DATA EFFICIENCY OBJECTIVES

We approximate the ground truth runtime $\mathcal{T}_a$ using learned cost models. Usually, these cost models are trained with one objective: to achieve high accuracy. However, due to the high cost of simulation in emerging hardware, we also want to minimize the amount of data samples required from these platforms for model training. We formalize these two objectives as follows.

**Data Collection Objective (DCE).** Let $D_a = \{(M_l, c_i), t_i \mid i \in m_a, l \in \mathbb{Z}^+\}$ be the dataset collected from hardware platform $a$, and let $\beta_a$ represent the average cost of collecting a single data sample from the platform. Our objective is to $\min_{D_a} \beta_a \times |D_a|$.

**Accuracy Objective.** Let $CM_a$ (which approximates $\mathcal{T}_a$) be the learned cost model trained on dataset $D_a$. If the best program configuration returned by the cost model ($c^*_{CM_a}$) has an actual execution time $t^*_{CM_a}$, our objective is to $\min |t^*_{CM_a} - t^*|$, where $t^*$ is the execution time for the optimal configuration. For a set of input sparse matrices $\{M_1, M_2, \ldots, M_k\}$, our objective can be extended to minimizing the Absolute Percentage Error (APE) across all matrices:

$$APE = \frac{1}{k} \sum_{l=1}^{k} \frac{|t^*_{CM_a, M_l} - t^*_{M_l}|}{t^*_{M_l}} \times 100$$

where $t^*_{CM_a, M_l}$ denotes the execution times for the predicted best program configuration for the input sparse matrix $M_l$ and $t^*_{M_l}$ denote the optimal program configurations for the same matrix.

## A.3 EVALUATIONS FOR COST MODEL OBJECTIVES

To evaluate the cost model objectives, we conducted the following experiments for SpMM on SPADE. For simplicity in the calculations, we set $\beta_{\text{CPU}} = 1$ and $\beta_{\text{SPADE}} = 1000$. However, a CPU execution typically takes milliseconds, while a SPADE execution can extend up to two weeks. We explored 11 distinct models across four different categories, differentiated by the number of data samples they were trained on, while the cost model architecture remained the same. Category I consists of models (NT $d$) trained exclusively on data samples from $d$ matrices executed on SPADE. Category II includes transfer-learned models (TL $d$), which were pre-trained with data samples from 100 matrices on CPU (10,000 data samples) and then fine-tuned on SPADE with data samples from $d$ matrices. Category III consists of models (CPU $d$) pre-trained with varying numbers of data samples from $d$ matrices on CPU and then fine-tuned on data samples from 5 matrices on SPADE. Finally, we did zero-shot inference (Zero-Shot) from a model pre-trained on CPU with data samples from 100 matrices without additional fine-tuning on SPADE.

Models trained exclusively on SPADE data samples (NT d) generally exhibit increasing speedup and decreasing APE as the number of SPADE data samples increases. For example, NT 1000, trained on 100,000 SPADE data samples, achieves the highest speedup of 1.43 and an APE of 7.06. However, the data collection overhead for these models rises significantly with the number of SPADE samples, making the use of them impractical due to the long simulation times. In contrast, the TL models, which are pre-trained on CPU data and fine-tuned on SPADE samples, demonstrate an excellent balance between speedup, APE, and DCE. TL 5 model, for instance, delivers a competitive speedup of 1.40 and a low APE of 7.28, while maintaining an excellent DCE of 0.51.

| Model | Data Samples | | Dash (Top-1) Speedup | APE | DCE$/10^6$ |
|---|---|---|---|---|---|
| | CPU | SPADE | | | |
| NT 5 | - | 500 | 1.29 | 15.02 | 0.50 |
| NT 100 | - | 10000 | 1.38 | 9.42 | 10.00 |
| NT 1000 | - | 100000 | 1.43 | 7.06 | 100.00 |
| TL 5 (CPU 100) | 10000 | 500 | 1.40 | 9.58 | 0.51 |
| TL 100 | 10000 | 10000 | 1.41 | 8.74 | 10.01 |
| TL 1000 | 10000 | 100000 | 1.42 | 7.28 | 100.01 |
| CPU 5 | 500 | 500 | 1.07 | 27.80 | 0.50 |
| CPU 20 | 2000 | 500 | 1.21 | 19.35 | 0.50 |
| CPU 500 | 50000 | 500 | 1.36 | 16.34 | 0.55 |
| CPU 1000 | 100000 | 500 | 1.19 | 36.00 | 0.60 |
| Zero-Shot (CPU) | 10000 | - | 0.71 | 46.22 | 0.01 |

Table 2: Comparison of cost model performance with varying data samples from CPU and SPADE.

## A.4 LEARNING OBJECTIVE

Our objective is to train a cost model that effectively learns to identify a program configuration that minimizes the runtime of a sparse operation for a given sparsity pattern. To achieve this, we begin by training our cost model to learn the relative rankings of program configurations during execution, enabling it to accurately identify optimal configurations based on their performance. This allows us to combine our cost model with a search technique to efficiently select the top-k (k-best) program configurations from the configuration space. We use the *pairwise ranking loss* as our learning objective (implemented using margin ranking loss) to rank program configurations based on their true performance differences. For a given input matrix $M$, the *pairwise ranking loss* $(\mathcal{L})$ across all program configuration pairs can be defined as $\mathcal{L} = \sum_{(c_1,c_2)} \max(0, 1 - (\hat{r}_{M,c_1} - \hat{r}_{M,c_2})) \cdot \delta_{\text{true}}$; $\delta_{\text{true}} = \text{sign}(t_{M,c_1} - t_{M,c_2})$ where $\hat{r}_{M,c_1}$ and $\hat{r}_{M,c_2}$ are the predicted scores for configurations $c_1$ and $c_2$, respectively; $t_{M,c_1}$ and $t_{M,c_2}$ represent their actual runtimes; and $\delta_{\text{true}}$ signifies the true performance difference where $\text{sign}(x)$ returns 1 if $x > 0$, -1 if $x < 0$, and 0 if $x = 0$. This ensures that the model is penalized when the predicted ranking does not align with the true ranking. By minimizing this loss $(\mathcal{L})$, DASH improves its ability to accurately rank and identify the top-k program configurations. This also contributes to achieving our *accuracy objective* (Section A.2).

## B CODE OPTIMIZATIONS ACROSS HARDWARE PLATFORMS

- Loop strip-mining: Breaks down large software loops into smaller segments to optimize cache utilization.
- Loop reordering: Adjusts the execution order of loops to improve cache efficiency. Typically, it is applied after loop strip-mining.
- Format reordering: Reorganizes the data structure layout of sparsity patterns in memory to optimize memory access patterns
- Parallelization: Distributes tensor computations across multiple threads or processors to run tasks simultaneously.
- Loop binding: Assigns specific loop iterations to threads for parallel processing.
- Loop unrolling: Executes multiple iterations of a loop in a single iteration, reducing loop control overhead and boosting execution speed.

- Tiling: Decomposes a matrix into smaller blocks to optimize data reuse in the local memory and improve cache efficiency.

- Barrier: Applying a barrier would ensure all threads finish processing their current tile (synchronized) before progressing to the next stage.

- Cache bypassing: Capability of bypassing caches to to reduce cache pollution.

- Matrix reordering: Enhances data locality by reordering the input matrix.

## C HYPERPARAMTERS

Table 3: Hyperparameters for model training/fine-tuning

| Hyperparameter | Value |
|---|---|
| Learning Rate | 0.0001 |
| Batch Size | 32 |
| Optimizer | Adam |
| Number of Epochs | 100 |
| Loss Function | MarginRankingLoss |

Table 4: Composition of layers in the Input Featurizer ($\mathcal{IFE}$)

| Layer | Description |
|---|---|
| Layer 1 | MinkowskiConvolution (in_channels, 32, kernel_size=5) |
| Layer 2 | MinkowskiConvolution (32, 32, kernel_size=3) |
| Layer 3 | MinkowskiConvolution (32, 64, kernel_size=3) MinkowskiMaxPooling |
| Layer 4 | MinkowskiConvolution (64, 64, kernel_size=3) |
| Layer 5 | MinkowskiConvolution (64, 64, kernel_size=3) |
| Layer 6 | MinkowskiConvolution (64, 128, kernel_size=3) MinkowskiMaxPooling |
| Layer 7 | MinkowskiConvolution (128, 128, kernel_size=3) |
| Layer 8 | MinkowskiConvolution (128, 128, kernel_size=3) |
| Layer 9 | MinkowskiConvolution (128, 256, kernel_size=3)MinkowskiMaxPooling |
| Layer 10 | MinkowskiConvolution (256, 256, kernel_size=3) |
| Layer 11 | MinkowskiConvolution (256, 256, kernel_size=3) |
| Layer 12 | MinkowskiConvolution (256, 256, kernel_size=3) |
| Global Pooling Layer | MinkowskiGlobalAvgPooling |

Table 5: Composition of layers in the Predictor ($\mathcal{P}$)

| Component/Layer | Input Size | Output Size |
|---|---|---|
| Matrix Embedding (x) | 128 | 128 |
| Configuration Embedding (y) | 53 | 64 |
| Latent Embedding (z) | 64 | 64 |
| Concatenation (xyz) | 128 + 64 | 192 |
| Predictor Layer 1 | 192 | 128 |
| Predictor Layer 2 | 128 | 64 |
| Predictor Layer 3 | 64 | 1 |

# D    ADDITIONAL RESULTS

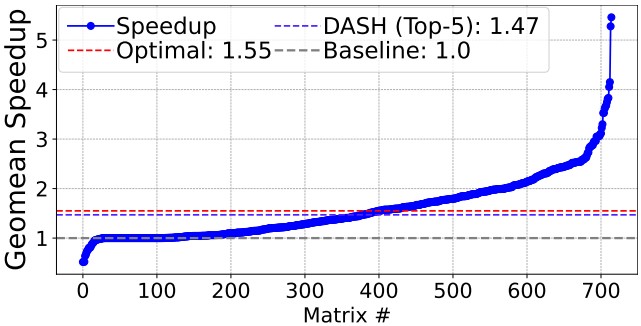

Figure 13: DASH (Top-5) per-matrix speedups (SpMM)

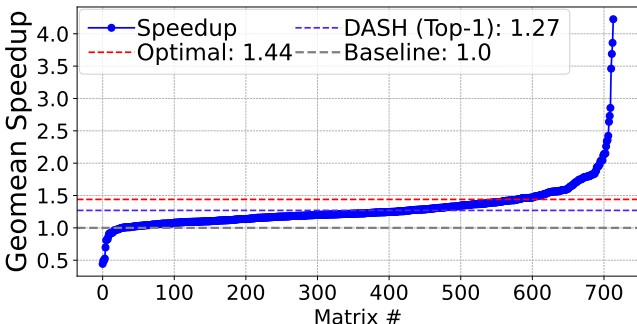

Figure 14: DASH (Top-1) per-matrix speedups (SDDMM)

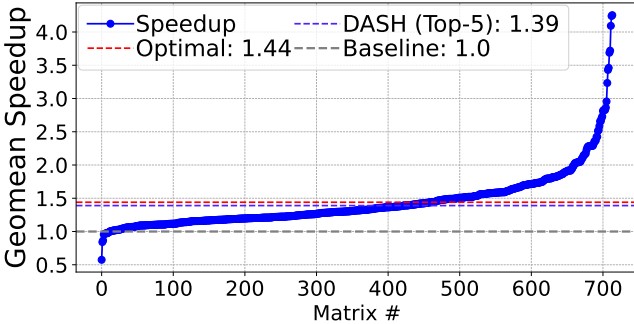

Figure 15: DASH (Top-5) per-matrix speedups (SDDMM)

