# OpenReview forum: "DASH: Data-Efficient Learned Cost Models for Sparse Matrix Computations on Emerging Hardware Platforms"
_ICLR.cc/2025/Conference — Submitted to ICLR 2025_

### Official Review · Reviewer_5VNi · 2024-10-22

**Soundness:** 2
**Presentation:** 3
**Contribution:** 2
**Rating:** 6
**Confidence:** 3

**Summary:**

Cost models are used to predict the performance of kernel programs so that more efficient kernel programs will be used to perform the computation. This papers targets at building data-efficient learned cost model for emerging hardwares by transfer learning, which requires less data samples from the new hardware that might only be only avaliable as simulators in the early stage of hardware development. The authors propose a few-shot fine-tuning technique, called DASH, to first train a cost model on widely used hardware and then fine-tune the model on the target hardware with a small number of data samples. The cost model considered the spare matrix pattern, homogenous and heterogenous kernel program configurations so that it can be trained and fine-tuned for different hardwares. Experiments show that DASH outperforms existing techniques that use transfer learning by 28.44% interms of the execution performance of found kernels on the target hardware.

**Strengths:**

1. A cost model architecture design that considered the heterogeneity of hardware so that it can be trained and fine-tuned for different hardware, enabling the transfer learning.
2. The paper is well written and easy to follow.

**Weaknesses:**

1. The motivation to have such a cost-model for an emerging hardware at the very eary stage is weak.
2. The avaliablility of a good representation of kernel program configuration for emerging hardware is questionable.
3. The method is highly coupled with the underlying kernel generator.

See below comments for more details.

**Questions:**

1. The motivation for developing a cost model for emerging hardware at such an early stage is weak. The main motivation for this work is that we can only obtain a very small set of data samples to train the cost model; thus, we can use transfer learning to reduce the demand for a large number of data samples on the target hardware. However, why do we need such a cost model at such an early stage when the hardware development is still in its infancy, making it difficult to obtain sufficient data samples? The common use of a cost model in existing hardware (CPUs, GPUs, and TPUs) is to help select the best kernel from a group of kernels that is optimal for a given input and hardware. The authors claim that such a cost model can assist hardware designers in determining the optimal performance they can achieve at the design stage. However, at this early stage, a good profiler or simulator is more important than a cost model, as the hardware designer can continuously optimize the kernel (which may be written in assembly code or low-level programming) based on feedback from the profiler and simulator. At such an early stage, it is highly questionable whether there is a reliable compiler or kernel generator that can abstract the kernel into a set of tunable parameters (see point 3).

2. The availability of a good representation of kernel program configurations for emerging hardware is questionable. The proposed cost model requires that the optimizations in the kernel programs on the emerging hardware can be abstracted into a set of predefined configurations. However, this greatly limits the scope of the kernels supported by the cost model. Even for mature hardware like NVIDIA GPUs, kernels can be implemented in CUDA in various ways, making it difficult to model CUDA kernels directly. If we restrict the scope of supported kernels, we might miss many efficient kernels during kernel searching.

3. The method is highly coupled with the underlying kernel generator. DASH relies on a high-level kernel composer/generator to create the kernels, which restricts DASH to predicting only the kernel performance generated by those specific kernel generators. In the paper, DASH relies on TACO to generate CPU kernels and SparseTIR to generate NVIDIA GPU kernels. Therefore, the new hardware must have a similar high-level kernel composer to adopt DASH.

---

> ### Author Response · Authors · 2024-11-20
> **Response to Reviewer 5VNi (part 1)**
>
> We sincerely thank you for the time and effort you dedicated to carefully reviewing our work and for providing valuable and constructive feedback. We especially want to thank you for accurately summarizing our paper’s contributions. In the following responses, we have carefully addressed each of the concerns you raised.
>
> > ***Weakness 01 & Question 01*** \
> > The motivation for developing a cost model for emerging hardware at such an early stage is weak.
>
> **Response:** Thank you very much for raising this important concern. We agree that a good profiler and a simulators are important for a hardware architect. However, for accelerators specifically designed for complex applications like sparse matrix computation, an accurate simulator would take a significant time to execute. Further, the increased configurability of sparse accelerators during recent years has made ***Design Space Exploration (DSE)*** significantly more complex as sparse operation (kernels) must be carefully optimized to fully leverage the underlying hardware's performance.
>
> This has led architects to face the dual challenge of evaluating optimal hardware choices (e.g., cache sizes, memory hierarchies, and compute units) while also determining the best software (program configurations) tailored to specific input characteristics for a given sparse operation (kernel). Hence, in the presence of any inefficiencies observed during early-stage simulations, it has become difficult to diagnose whether the inefficiencies stem from suboptimal hardware choices or unoptimized software.
>
> Architects don’t want to construct hardware that is overprovisioned (e.g., increase the sizes of caches) if an inefficiency could instead be tackled with better software (e.g. a different tiling strategy). Hence,  it is crucial to address these issues during the early design stages, as making changes to the hardware becomes exponentially more expensive as the chip design progresses. One way to tackle this is to exhaustively evaluate (in the simulator) all possible software choices for every different sparse input and hardware configuration. However, due to the high cost of simulations, this becomes infeasible. For every design point, running simulations across all possible inputs and program configurations would require an impractical amount of computational resources and time.
>
> Conversely, a cost model offers a practical solution to this challenge by delivering fast and accurate performance predictions across a wide range of program configurations, thereby significantly reducing the reliance on extensive and computationally expensive simulations. While simulators and profilers remain essential, a well-designed cost model complements these tools by streamlining the design process and enabling efficient exploration of the design space, especially when transfer learning is employed to leverage limited data samples effectively. This synergy between simulators, profilers, and cost models accelerates making good design decisions, paving the way for the development of high-performance, cost-efficient accelerators. **In summary, DASH removes a dimension of complexity from the DSE stage by automatically selecting the best software program configuration for a given input and hardware configuration.**

---

> > ### Author Response · Authors · 2024-11-20
> > **Response to Reviewer 5VNi (part 2)**
> >
> > > ***Weakness 02 & Question 02*** \
> > > The availability of a good representation of kernel program configurations for emerging hardware is questionable.
> >
> > **Response:** The concern about representing kernel program configurations for GPUs is valid, and we acknowledge that some accelerators may fall outside the scope of our abstraction. However, our approach is highly effective for accelerators that implement a predefined set of kernels (e.g., SPMM, SDDMM, etc.) with rich tunable parameters (e.g., tile size). Unlike GPUs, which are highly programmable with a vast implementation space, accelerators are typically domain-specific and highly parameterized. Thus, our abstraction based on predefined configurations with tunable parameters is sufficient to describe the optimization space for many sparse accelerators. For instance, the SPADE paper [1] includes such a well-defined set that aligns well within with our abstraction. To elaborate, there is no possible SPADE execution mode that our abstraction does not capture.
> >
> > > ***Weakness 03 & Question 03*** \
> > > DASH relies on a high-level kernel composer/generator to create the kernels.
> >
> > **Response:** The concern about DASH's reliance on specific kernel generators, such as TACO for CPU kernels and SparseTIR for NVIDIA GPU kernels, is true. However, we do not require a kernel composer to generate kernels in accelerators. **During the early design stage, we can use a simple configuration file to modify and tune the parameters of the kernel**, eliminating the need for a complex kernel composer. For example in SPADE, the kernel space is limited to a few core kernels, such as SpMM and SDDMM. These kernels are highly parameterized, allowing us to represent and explore their configurations systematically without requiring a full-fledged kernel generator.
> >
> > [1] Gerogiannis, Gerasimos, et al. "Spade: A flexible and scalable accelerator for spmm and sddmm." Proceedings of the 50th Annual International Symposium on Computer Architecture. 2023.

---

> > > ### Comment · Reviewer_5VNi · 2024-11-24
> > >
> > > Thank you to the authors for providing detailed and thoughtful responses to the concerns I raised in my review. I truly appreciate the effort put into addressing these points.
> > >
> > > That said, I still believe that relying on experts and a high-performance simulator is the most effective approach for implementing efficient kernel design in the early stages of chip design. Typically, chips are designed with a series of target applications in mind, allowing the most critical workloads to serve as a baseline for optimization. Importantly, the number of significant workloads is not unlimited.
> > >
> > > In contrast, incorporating a cost model introduces additional complexity to the design process. Engineers would need to adapt the cost model to the new hardware, which could restrict design exploration if the model does not account for novel features. Alternatively, enhancing the cost model to support new features can be equally challenging. By directly relying on kernel experts with hardware expertise, these issues—and the ones I mentioned in my review—can be avoided entirely.
> > >
> > > With this perspective in mind, I tend to maintain my original score. Nonetheless, I want to sincerely thank the authors for their valuable contributions to the methodological advancements and their detailed responses.

---

> > > > ### Author Response · Authors · 2024-11-26
> > > > **Response to Reviewer 5VNi (part 3)**
> > > >
> > > > We sincerely thank the reviewer for your detailed feedback and for taking the time to engage with our responses. We truly appreciate the acknowledgment of our contributions in this work and our efforts in addressing the concerns raised. We respect your decision to maintain your original score, and understand and respect your perspectives. Below, we have addressed the concerned raised in your feedback. Please feel free to let us know if any aspects require further clarification. We remain open to any additional discussions you may have.
> > > >
> > > > > ***Concern 01*** \
> > > > > I still believe that relying on experts and a high-performance simulator is the most effective approach for implementing efficient kernel design in the early stages of chip design.
> > > >
> > > > **Response:** We agree that relying on expert-driven approaches is effective and has a significant role in the early stages of chip design. However, with the flexibility of recent accelerators to have software programmable kernels [1,2,3], integration of cost models and heuristics into the DSE pipeline has become an up and coming area [1,4]. For example, Vesper is a recent work that had integrated an analytical model to a configurable sparse accelerator to enable higher throughput [1]. HotTiles is another work that uses an analytical model to predict the performance of different accelerator processing elements (PEs) that accommodate intra-matrix heterogeneity [4]. Further, in HotTiles, the authors acknowledged that a more accurate model could have enabled making better design decisions during the early stages. We believe that our proposed data-driven cost model framework, DASH, addresses this gap (resulting in speedups close to optimal (Figure 7)) while complementing expert-driven strategies to enable more informed and better design decisions.
> > > >
> > > > > ***Concern 02*** \
> > > > > Typically, chips are designed with a series of target applications in mind, allowing the most critical workloads to serve as a baseline for optimization. Importantly, the number of significant workloads is not unlimited.
> > > >
> > > > **Response:** We agree with the reviewer that this observation holds true for CPU benchmark suites like SPEC, which comprises of 43 benchmarks. In contrast, the corresponding benchmark suite for sparse applications, SuiteSparse, contains 2893 benchmarks, a significantly larger set. This difference arises due to the presence of diverse sparsity patterns in sparse matrices. By automating software configuration selection, we can evaluate a larger subset of these benchmarks, enabling the design of a chip that accommodates a broader range of sparsity patterns. Further, the time saved could be redirected to exploring more DSE points, such as testing a wider range of cache sizes.
> > > >
> > > > > ***Concern 03*** \
> > > > > In contrast, incorporating a cost model introduces additional complexity to the design process.
> > > >
> > > > **Response:** As mentioned earlier, in the context of configurable accelerators, there is a growing trend to rely on either heuristic or analytical cost models to select software configurations [1,4]. We propose replacing these methods with a data driven approach. The primary overhead associated with our approach arises from the need to gather data points to fine-tune the cost model. This overhead is minimal compared to the effort required for an expert to iteratively optimize a kernel for sparse workloads, where kernel performance is highly input-sensitive due to diverse sparsity patterns.
> > > >
> > > > > ***Concern 04*** \
> > > > > Enhancing the cost model to support new features can be equally challenging.
> > > >
> > > > **Response:** In our cost model architecture, hardware-specific optimization parameters (features) are  automatically included in the heterogeneous component using the latent encoder. Hence, any additional features related to new hardware-specific optimizations can be easily integrated to the cost model with minimal effort during fine-tuning.
> > > >
> > > > Thank you once again for your valuable insights and feedback. We will definitely take your perspective into account as we continue to develop this work.
> > > >
> > > > [1] Jin, Hanchen, et al. "Vesper: A Versatile Sparse Linear Algebra Accelerator With Configurable Compute Patterns." IEEE Transactions on Computer-Aided Design of Integrated Circuits and Systems (2024).
> > > >
> > > > [2] Gerogiannis, Gerasimos, et al. "Spade: A flexible and scalable accelerator for spmm and sddmm." Proceedings of the 50th Annual International Symposium on Computer Architecture. 2023.
> > > >
> > > > [3] Muñoz-Martínez, Francisco, et al. "Flexagon: A multi-dataflow sparse-sparse matrix multiplication accelerator for efficient dnn processing." Proceedings of the 28th ACM International Conference on Architectural Support for Programming Languages and Operating Systems, Volume 3. 2023.
> > > >
> > > > [4] Gerogiannis, Gerasimos, et al. "HotTiles: Accelerating SpMM with Heterogeneous Accelerator Architectures." 2024 IEEE International Symposium on High-Performance Computer Architecture (HPCA). IEEE, 2024.

---

> > > > > ### Comment · Reviewer_5VNi · 2024-11-27
> > > > >
> > > > > Thank authors for the latest response, which partially addressed my concerns, especially on the motivation parts, raising the score from 5 to 6.

---

> > > > > > ### Author Response · Authors · 2024-11-28
> > > > > > **Response to Reviewer 5VNi (part 4)**
> > > > > >
> > > > > > We sincerely thank Reviewer 5VNi for raising the score. We are pleased to hear that our clarifications have addressed some of your concerns. Your comments and suggestions have been extremely helpful in revising our paper, and we will take them into account as we continue to develop this work. Thank you again for your time and constructive feedback.

---

### Official Review · Reviewer_bgES · 2024-11-02

**Soundness:** 3
**Presentation:** 3
**Contribution:** 3
**Rating:** 6
**Confidence:** 3

**Summary:**

This paper presents DASH, a framework for developing data-efficient learned cost models for sparse matrix computations on emerging hardware platforms. The key innovation is enabling effective transfer learning from widely available general-purpose hardware (e.g. CPUs) to emerging accelerator platforms that only have access to expensive simulators during early design stages. DASH introduces techniques to handle both homogeneous and heterogeneous aspects of program configurations across platforms, using approximate mapping for comparable code optimizations and latent space encoding for hardware-specific optimizations. The authors evaluate DASH on SpMM and SDDMM, showing it can achieve comparable accuracy using only 5% of the data samples required by models trained exclusively on accelerator data.

**Strengths:**

* The wrok addresses an important problem of the high cost of collecting training data through simulation in hardware acceleration design.
* Novel technical approach of separating program configurations into homogeneous and heterogenous components and using autoencoders for compact latent representation of hardware-specific parameters.
* Evaluation shows strong results (speedups of 1.47x for SpMM, 1.39x for SDDMM)

**Weaknesses:**

* Limited theoretical analysis or justification for the design choices. For example, autoencoders are used for latent but lacking explanation for why this particular architecture was chosen over alternatives. An ablation study could help justify the chosen design.
* The evaluation and results focuses heavily on SPADE. Results on other accelerator architectures would strengthen the generalizability claims. While GPU results are included, they could benefit from more details analysis, e.g. comparison of transfer learning effectiveness between CPU->GPU vs. CPU->SPADE.

**Questions:**

1. How sensitive is DASH to the choice of autoencoder architecture? What other architectures were considered?
2. How do the results on GPU compare to cuSPARSE?
3. What are the limitations of the approximate mapping approach? What are the key assumptions about the similarity between the source and target platforms? Are there cases where it breaks down? Are there certain types of optimizations that cannot be effectively mapped?
4. What modifications would be needed to handle accelerators with significantly different architectures (from SPADE)? How much effort is it to find the approximate mapping for such architectures?

---

> ### Author Response · Authors · 2024-11-20
> **Response to Reviewer bgES (part 1)**
>
> We greatly appreciate the reviewer for dedicating the time and effort to providing thorough and constructive feedback. Furthermore, we are especially thankful to you for highlighting the importance of the problem we addressed and for recognizing the novelty of our work. In the following responses, we have carefully addressed each of your suggestions and concerns.
>
> > ***Weakness 01 & Question 01*** \
> > Limited theoretical analysis or justification for the design choices
>
> **Response:** In our initial exploration, we explored maximum mean discrepancy (MMD) as a potential alternative for heterogeneous transfer. However, this approach did not yield promising results. We also experimented with heterogeneous feature augmentation (Figure 2), which produced suboptimal outcomes. To justify our final choice for the design, we will include an ablation study in our revised submission.
>
> > ***Weakness 02*** \
> > Results on other accelerator architectures would strengthen the generalizability claims.
>
> **Response:** We agree that evaluations on multiple accelerator architectures would have strengthened the generalizability claims. Among the sparse accelerators we explored, we found SPADE to be the most challenging due to its exposure to a rich program configuration space. Further, it supports two sparse matrix operations (kernels), SpMM and SDDMM, in contrast to [1,2] (which only supports SpMSpM/SpGEMM). In our experiments, we considered the GPU as a good candidate to demonstrate generalizability due to its programmability, treating it as an architecture that is vastly different from SPADE. Further, in response to question 04, we will provide the intuition into implementing DASH on other sparse accelerators [1,3,4]. We anticipate a growing trend in the prevalence of sparse accelerators that integrate greater flexibility in their software program configurations where DASH can be applicable.
>
> > ***Question 02*** \
> > How do the results on GPU compare to cuSPARSE?
>
> **Response:** sparseTIR [5] had provided speedups for 6 different datasets where they were on par or better than cuSPARSE. However, considering the suggestion by the reviewer we evaluated against cuSparse for all 715 matrices using cusparseSpMM kernel. cusparseSpMM had a geo mean speedup of x1.0176 compared to our baseline. Our (DASH's) top-1 geo mean speedup for the same kernel was x1.0309 and the top-5 geo mean speedup was x1.1689.
>
> > ***Question 03*** \
> > What are the limitations of the approximate mapping approach? Are there cases where it breaks down?
>
> **Response:** Thank you very much for pointing this out. Our approximate mappings are designed to enable well-established code optimizations that remain effective across emerging hardware platforms. For instance, we expect loop transformations (e.g. loop strip-mining, loop reordering, tiling, etc) to be implemented regardless of the underlying hardware platform although the exact implementation may be slightly different. Any hardware-specific optimizations (code optimizations that cannot be mapped) are included in the heterogeneous component using the latent encoder. Since the dimensions of the latent embedding are fixed, we can effectively finetune using an already pre-trained model.
>
> Further,  if we encounter a hardware platform with code optimizations that differ significantly from a general-purpose hardware platform like a CPU, we would not be able to incorporate a homogenous component in the cost model. In such cases, retraining the cost model would be needed. In the absence of a homogeneous component, knowledge transfer will primarily rely on insights related to sparsity patterns, which we believe may lead to suboptimal results.
>
> [1] Muñoz-Martínez, Francisco, et al. "Flexagon: A multi-dataflow sparse-sparse matrix multiplication accelerator for efficient dnn processing." Proceedings of the 28th ACM International Conference on Architectural Support for Programming Languages and Operating Systems, Volume 3. 2023.
>
> [2] Li, Zhiyao, et al. "Spada: Accelerating sparse matrix multiplication with adaptive dataflow." Proceedings of the 28th ACM International Conference on Architectural Support for Programming Languages and Operating Systems, Volume 2. 2023.
>
> [3] Aananthakrishnan, Sriram, et al. "The Intel programmable and integrated unified memory architecture graph analytics processor." IEEE Micro 43.5 (2023): 78-87.
>
> [4] Gerogiannis, Gerasimos, et al. "HotTiles: Accelerating SpMM with Heterogeneous Accelerator Architectures." 2024 IEEE International Symposium on High-Performance Computer Architecture (HPCA). IEEE, 2024.
>
> [5] Ye, Zihao, et al. "Sparsetir: Composable abstractions for sparse compilation in deep learning." Proceedings of the 28th ACM International Conference on Architectural Support for Programming Languages and Operating Systems, Volume 3. 2023.

---

> > ### Author Response · Authors · 2024-11-20
> > **Response to Reviewer bgES (part 2)**
> >
> > > ***Question 04*** \
> > > How to handle accelerators with significantly different architectures (from SPADE)?
> >
> > **Response:** As we explained in our response to "Question 03", we expect loop transformations (e.g. loop strip-mining, loop reordering, tiling, etc) to be implemented regardless of the underlying hardware as they are well-established code optimizations considered for kernels like SpMM and SDDMM. To elaborate, let us explore the intuition behind incorporating approximate mapping of these loop transformations into sparse accelerators, using Intel PIUMA [3,4] and Flexagon [1] as examples. These mappings closely align with the approach we employed in SPADE.
> >
> > Intel PIUMA is a configurable accelerator that has a RISC ISA making it CPU-programmable. This enables it to employ code optimizations that are available in CPUs. Hence, it is possible to implement SpMM and SDDMM kernels with code optimizations such as loop reordering with a one-to-one mapping. Similarly, loop strip-ming and tiling can be mapped. However, similar to SPADE, where we accounted for the "barrier" optimization in the mapping process, we need to consider the scratchpad reuse here.
> >
> > Flexagon is another reconfigurable accelerator for sparse computations. Flexagon’s configurations can be mapped to loop reordering as three different data flow models use three loop traversal orders. Further, we can consider the usage of different formats of different matrices in Flexagon as a heterogeneous feature for the hardware platform. Although Flexagon has mentioned that they are using “tiling”, they have not provided a detailed implementation of it (source code is not available) or how tile size selection is done. However, considering standard tiling implementation, we are confident that we can represent Flexagon’s tiling mechanism using loop-stripping and loop reordering.
> >
> > Unfortunately, Intel PIUMA is proprietary, and Flexagon’s source code was not available, so we were unable to test our hypothesis on these accelerators. SPADE was the only accelerator we were able to gain access to, and we used it for our evaluations to demonstrate that heterogeneous transfer can be effectively achieved in the domain of sparse accelerators using few-shot learning.

---

> > ### Comment · Reviewer_bgES · 2024-11-25
> >
> > Thank you for taking the time to provide a thorough and thoughtful response to the points I raised in my initial review.
> >
> > The additional context and justifications around the design choices, experimental setup, and potential extensibility to other accelerator architectures were helpful in better understanding the contributions and generalizability of the work. I look forward to seeing the ablation study and more detailed comparison to cuSPARSE in the revision, as I believe they will strengthen the paper.
> >
> > I also found the discussion on adapting the approximate mapping approach to accelerators like Intel PIUMA and Flexagon insightful. While I understand the challenges in actually testing on those proprietary architectures, the intuition you provided does give some confidence that the techniques could potentially be generalized.
> >
> > That said, I still have some reservations about the overall impact and practicality of the approach in real-world accelerator design workflows, as alluded to in my original review. As such, I will be maintaining my original scores.

---

> > > ### Author Response · Authors · 2024-12-02
> > > **Response to Reviewer bgES (part 3)**
> > >
> > > Thank you very much for your thoughtful and constructive feedback. We are glad that the additional context and justifications on our design choices, experimental setup, and extensibility provided a clearer understanding of our contributions. We have included additional experimental results in the revised paper to address the concerns in *Question 01* and *Question 02*. We have also strengthened our introduction to more effectively convey the overall impact and practicality of the approach.

---

### Official Review · Reviewer_rzMd · 2024-11-04

**Soundness:** 3
**Presentation:** 4
**Contribution:** 3
**Rating:** 6
**Confidence:** 3

**Summary:**

The authors introduce DASH to address the sparse matrix computations when facing different input sparsity patterns and code optimizations.
DASH trains learned cost models with four components, including configuration mapper, input featurizer, latent encoder, and predictor.
From DASH, low-cost data samples from widely accessible general-purpose hardware, followed by few-shot fine-tuning to efficiently adapt to emerging hardware platforms.
DASH introduces a novel approach that leverages the homogeneity of input features across different hardware platforms while effectively mitigating heterogeneity.
From the evaluation, DASH can significantly outperforms existing techniques. By avoiding redundant computations and reducing optimization time, combined with the adaptability of deep learning-enhanced models, DASH paves the way for substantial hardware performance improvements.
In conclusion, the paper presents a novel framework, that could help us better understand the challenges coming from Sparse Matrix-Matrix Multiplication and Sampled Dense-Dense Matrix Multiplication. They develop learned cost models which can be leveraged across different hardware platforms.

**Strengths:**

* The idea from DASH is interesting. Utilizing technologies like autoencoders, DASH successfully mitigates the issue of negative transfer in heterogeneous feature spaces, outperforming traditional manual feature engineering and cost modeling methods. This approach effectively bridges different hardware platforms such as CPUs, GPUs, and sparse accelerators through heterogeneous transfer learning, making it an intuitive and efficient solution.
* DASH is not only applicable to sparse accelerators but also efficiently migrates to various hardware platforms, including GPUs, demonstrating robust generalization capabilities. Based on the WACO model, and by integrating statistical information capture with transfer learning, DASH flexibly extends to different hardware and complex computational tasks, reducing dependency on specific hardware and ensuring good adaptability for future new hardware.
* The paper has good coherence, and is well-structured. The background section explains the related work in an easy understandable way.
* The paper is also very clear with thorough experiments and analysis.

**Weaknesses:**

* The first concern arises from the insufficient coverage of sparse matrix structures and different operations. There exists a variety of sparse matrix structures employed in training data configurations, which are not fully demonstrated. The widely used sparse storage formats such as CSR, COO, CSC, ELL, DIA, BSR, and BELL are highly diverse. When dealing with the same type of sparse matrix, the computational patterns of optimization operations like SpMV and SpMM can be quite varied. They exhibit different sensitivities to matrix structures. Therefore, the fine-tuning needs to consider both the sparse matrix representation, the computational operation, and the optimization strategy. However, the performance of DASH solely relies on the types of sparse matrices and computational patterns covered during training. The model may not immediately provide optimal optimizations for novel sparse matrix structures or computational requirements not encountered in the training data, necessitating further training or fine-tuning. Please clarify this point.
* Th second concern arise from limited breadth of experimental evaluation, when dealing with different emerging hardware accelerators. While DASH has demonstrated significant performance improvements in experiments, the testing platforms were primarily focused on specific sparse matrix accelerators, such as SPADE. Other emerging hardware was not included. The results presented show the migration effects from CPU to SPADE accelerators and from CPU to GPU, but there is a lack of consideration and evaluation of a more extensive range of emerging hardware platforms. Different sparse accelerators may have very distinct hardware architectures, and may limite the generalization ability of DASH’s transfer learning optimizations, is not yet sufficiently clear.
* In heterogeneous scenarios, the feature mapping of autoencoders may not be sufficient to capture all behaviors, and the use of a simple MLP predictor could be inadequate to handle complex hardware feature variations. The authors can add more evidences to prove this point.
* Although transfer learning reduces initial simulation costs, the frequent changes in emerging hardware may require timely updates to configuration mappings and fine-tuning, which can increase maintenance complexity.

**Questions:**

* Please see the weakness section.

**Details Of Ethics Concerns:**

None.

---

> ### Author Response · Authors · 2024-11-20
> **Response to Reviewer rzMd (part 1)**
>
> We are grateful to the reviewer for recognizing the novelty and effectiveness of our proposed method and, we sincerely appreciate the recognition of our comprehensive experimental results and analysis, as well as the acknowledgment that the paper is well-structured. Further, we especially want to thank you for taking the time and effort to provide a detailed summary of our paper’s contributions and strengths. In the following responses, we have carefully addressed each of your suggestions and concerns.
>
> > ***Weakness 01*** \
> > Insufficient coverage of sparse matrix structures and different operations
>
> **Response:** During model pre-training, we selected sparse matrices randomly while ensuring a balanced representation of their dimensions and sparsity (Section 4.2) from the SuistSparse Matrix Collection. Our test set (715 matrices) consists of a diverse representation of sparsity patterns and evaluation results show that our model has performed well for the majority of the test matrices (Figure 5).
>
> >  The widely used sparse storage formats such as CSR, COO, CSC, ELL, DIA, BSR, and BELL are highly diverse.
>
> **Response:** As you have correctly pointed out, our cost model is not optimized for the storage formats (e.g., CSR, COO, CSC, ELL, DIA, BSR, BELL) for sparse accelerators. However, this limitation does not affect DASH's primary objective, which is to demonstrate the feasibility of knowledge transfer in sparse computations across diverse hardware platforms. A potential solution to address this limitation is by extending DASH to incorporate the storage format as an additional parameter in the cost model. By treating the storage format as a potential optimization, DASH could predict the optimal storage format for a given input sparse matrix and sparse operation, along with other tunable parameters. We identified prior work [1] that applied a similar approach in other studies, and we view this as a potential extension of our contributions. Again, we are thankful to the reviewer for this suggestion.
>
> > ***Weakness 02*** \
> > Lack of consideration and evaluation of a more extensive range of emerging hardware platforms
>
> **Response:** Our current focus is on demonstrating the potential of DASH specifically for sparse computations, which is why we targeted a sparse accelerator like SPADE in our experimental evaluations. The goal was to showcase how DASH's transfer learning-based cost model (which leverages homogeneity and mitigates heterogeneity) can significantly enhance performance for these types of accelerators during the early design stages. However, the methodology we propose is not restricted to SPADE or even sparse accelerators. In this work, we considered GPU as a distinct hardware architecture to demonstrate generalizability considering GPU’s programmability and significant differences with SPADE. In our future work, DASH has the potential to be extended to other types of accelerators (outside the scope of this work since we focus on sparse computations) with similar parameterized optimization spaces [2].
>
> > ***Weakness 03*** \
> > Autoencoders may not be sufficient to capture all behaviors
>
> **Response:** The main focus of this work is to successfully leverage the concept of using homogeneous and heterogeneous components of program configurations to achieve the benefits of knowledge transfer in the domain of sparse computations. For the current work, we found autoencoders to be sufficiently better compared to more conventional techniques like feature augmentation. We plan to provide an ablation study to justify our choice of autoencoders in our revision. In our future work, as we explore expanding DASH to accommodate more diverse accelerators (as explained in our previous response), we agree that we need to look further into more complex neural network architectures.
>
> > Use of a simple MLP predictor could be inadequate to handle complex hardware feature variations.
>
> **Response:** In this work, we adapted a hardware-specific cost model architecture (WACO cost model) to enable heterogeneous transfer while avoiding unnecessary complexity in the model design. To achieve this, we introduced/modified only the components (configuration mapper, latent encoder, input featurizer) critical for enabling this transfer, leaving other components unchanged to maintain simplicity. However, as you have suggested we acknowledge that alternative techniques could be explored for the model's final predictor instead of using an MLP. In our revision, we will include an ablation study to evaluate the feasibility of other potential techniques.

---

> > ### Author Response · Authors · 2024-11-20
> > **Response to Reviewer rzMd (part 2)**
> >
> > > ***Weakness 04*** \
> > > Frequent changes in emerging hardware may require timely updates to configuration mappings
> >
> > **Response:** Thank you very much for pointing this out. While it is true that frequent changes in emerging hardware may require updates to configuration mappings and fine-tuning, this challenge is significantly mitigated by our approach in DASH. As long as the entire kernel does not change or the newly introduced optimizations are heterogeneous, updating the mappings is relatively simple. However, relying solely on simulations would require rerunning them for a large number of configurations each time a change is made, resulting in significant computational and time costs. In contrast, our transfer learning-based approach significantly reduces the cost and time of running simulations. By collecting only a few data samples and fine-tuning the model, we can efficiently adapt to hardware changes without the need for extensive simulations. Hence, this approach not only reduces maintenance complexity but also accelerates the design process, making it more feasible to handle frequent and timely updates in emerging hardware scenarios.
> >
> > [1] Zhao, Yue, et al. "Bridging the gap between deep learning and sparse matrix format selection." Proceedings of the 23rd ACM SIGPLAN symposium on principles and practice of parallel programming. 2018.
> >
> > [2] Kaufman, Sam, et al. "A Learned Performance Model for Tensor Processing Units." Proceedings of Machine Learning and Systems. 2021. 387-400.

---

> > > ### Comment · Reviewer_rzMd · 2024-11-25
> > >
> > > Thank you for your detailed response and clarifications. I genuinely appreciate the effort you've made to address my concerns. The context you provided was helpful in deepening my understanding of the contributions. However, my primary concern about the rationale for employing a uniform cost-model for different matrix representations and operations on emerging, large-scale hardware remains unaddressed. As such, I will maintain my original scores.

---

> ### Author Response · Authors · 2024-12-02
> **Response to Reviewer rzMd (part 3)**
>
> Thank you very much for your thoughtful and constructive feedback. We are glad that the additional context helped clarify our contributions. As you suggested, we have conducted ablation studies on the technique to capture the heterogeneity of program configurations (autoencoders) and the selection of the predictor  (*MLP predictor*)  and have included the results in the revised version of the paper. Further, as mentioned in our response to *Weakness 01*, we will consider the use of different sparse formats as a potential extension of our contributions.

---

### Official Review · Reviewer_R7XU · 2024-11-04

**Soundness:** 2
**Presentation:** 2
**Contribution:** 2
**Rating:** 3
**Confidence:** 3

**Summary:**

This paper presents DASH, a machine learning-based framework for constructing performance predictors to guide program transformations (e.g., loop-level optimizations) for sparse matrix computations on hardware platforms. DASH builds upon WACO, an existing ML framework tailored for sparse matrix computations, with new changes to WACO. Specifically, DASH uses a deeper neural network and maps program transformations into an embedding space to predict the execution time for a specific program transformation configuration on given input matrices and underlying hardware. DASH also leverages transfer learning to reduce the data collection overhead when adapting the trained cost model to new hardware platforms. Experimental results show that DASH delivers a higher accuracy in predicting the execution time, which leads to an overall better speedup.

**Strengths:**

The paper targets an important problem
Some promising results

**Weaknesses:**

A large part of the work relies on either strong user intervention (e.g., crafting approximate mapping of code optimizations) or existing techniques like deeper neural networks, mapping program transformation configurations into an embedding space, or transfer learning. The overall technical novelty appears to be rather limited.

The evaluation was performed on the sparse computation kernel level using matrices from the SuiteSparse Matrix Collection. It would be useful to show the end-to-end performance improvement of a realistic program (e.g., a graph neural network) and the prediction overhead w.r.t. to the end-to-end performance improvement.

The paper is difficult to follow at times, overly complicating simple things with math equations and formal definitions. Some of the details were not clear until reading the appendix. Important details are also missing. For example, as SparseTIR builds upon TVM, do you perform TVM schedule searching per matrix input? If so, did you include the search overhead?

**Questions:**

- Can you comment on the portability of the approach as it requires manually defining the mapping of code optimizations across different hardware?

- DASH benefits from a deeper DNN, mapping the code transformation configurations into an embedding space and transfer learning. These all seem to be standard ML techniques. What are the specific challenges for applying these techniques to build cost models for sparse matrix computation?

-What exactly are the default optimization used by TACO and SparseTIR as the baseline?

-Did you use parallelization in the evaluation? In other words, does the code run in parallel using multiple threads on multi-core CPUs?

---

> ### Author Response · Authors · 2024-11-20
> **Response to Reviewer R7XU (part 1)**
>
> We sincerely appreciate the time and effort the reviewer has invested in thoroughly reviewing our work and providing constructive insights. We are also truly grateful for your recognition of the importance of the problem we are addressing. In this context, our main contribution is the integration of machine learning-based optimization techniques (through learned cost models) for emerging sparse accelerators, enabling hardware architects to make critical design choices (e.g., cache sizes, memory hierarchies, and compute units) in the early stages of development.
>
> Below, we made every effort to address the concerns you raised. Please feel free to let us know if any aspects require further clarification. We remain open to any additional discussions you may have.
>
> > ***Weakness 01*** \
> > A large part of the work relies on strong user intervention (e.g., crafting approximate mapping of code optimizations)
>
> **Response:** Thank you very much for bringing this to our attention. As you have correctly pointed out, the approximate mappings of code optimizations were done with user intervention. However, these mappings have a significant role in the effectiveness of our cost model [Figure 7]. The reason we moved in this direction was motivated by the suboptimal results we received from feature augmentation: a more conventional off-the-shelf technique that does not require any user intervention (Figure 2). By employing approximations for code optimizations that had a similar impact, we enabled feature reuse across various programming systems (TACO, SparseTIR, and SPADE ISA) spanning multiple hardware platforms (CPUs, GPUs, and SPADE accelerator) for effective knowledge transfer. We will further discuss the intuition behind these approximations for other sparse accelerators in our response to "Question 01".
>
> > A large part of the work relies on existing techniques like deeper neural networks
>
> **Response:** Our contributions lie in using concepts from deep neural networks to develop a cost model architecture capable of leveraging homogeneity (through approximate mappings of code optimizations) while mitigating heterogeneity (using latent representations via an autoencoder) of program configurations across different programming systems. This enabled us to achieve an average geo-mean speedup of 1.40× (for SpMM) using only a few data samples, while applying popular heterogeneous transfer learning techniques (illustrated in Figure 2) yielded significantly suboptimal results (1.09× or less). Through our innovations, we overcome the limitations of prior work in the domain, which achieved effective knowledge transfer only between hardware platforms of the same architecture by focusing primarily on the homogeneous aspects of program configurations.
>
> > ***Weakness 02*** \
> >  end-to-end performance improvement of a realistic program
>
> **Response:** Thank you for providing this suggestion. In this work, we focus exclusively on sparse kernels because the early-stage accelerators we are addressing are specifically designed for sparse workloads and depend on simulators to generate runtime numbers. Assuming the dense computations of a GNN are performed on a CPU/GPU, we can hypothetically provide end-to-end performance metrics for GCN training. Furthermore, for GPUs, where both sparse and dense computations are executed on the GPU, we will provide evaluations for the end-to-end performance of GCN training in comparison with the established DGL framework. We will include these additional experimental results in our revision.
>
> > ***Weakness 03*** \
> >  The paper is difficult to follow at times. Some of the details were not clear until reading the appendix.
>
> **Response:** Thank you very much for pointing this out. We will do our best in the rebuttal to address these concerns. What we wanted to convey by “SparseTIR was built on the TVM compiler” is that SparseTIR was developed as an enhancement to TVM's Tensor IR, building upon the broader TVM infrastructure. We recognize that this distinction may not have been sufficiently clear in the paper and will make sure to elaborate on it.

---

> > ### Comment · Reviewer_R7XU · 2024-11-25
> >
> > Thank you for your efforts in drafting the response. I have read through the response, but my main concerns still stand - the proposed approach relies heavy user intervention and the performance of the proposed appears to be largely depend on the capability of a deep neural network. Hence, I will keep my original score.

---

> ### Author Response · Authors · 2024-11-20
> **Response to Reviewer R7XU (part 2)**
>
> > ***Question 01*** \
> >  The portability of the approach as it requires manually defining the mapping of code optimizations across different hardware
>
> **Response:** Thank you very much for bringing this important question to our attention. Our approximate mappings are designed to enable well-established code optimizations that remain effective across emerging hardware platforms. For instance, we expect loop transformations (e.g. loop strip-mining, loop reordering, tiling, etc) to be implemented regardless of the underlying hardware platform although the exact implementation may be slightly different. Any hardware-specific optimizations (code optimizations that cannot be mapped) are included in the heterogeneous component using the latent encoder. Since the dimensions of the latent embedding is fixed, we can effectively finetune using an already pre-trained model. To elaborate, let us explore the intuition behind incorporating approximate mapping of these loop transformations into sparse accelerators, using Intel PIUMA [1,2] and Flexagon [3] as examples. These mappings closely align with the approach we employed in SPADE.
>
> Intel PIUMA is a configurable accelerator that has a RISC ISA making it CPU-programmable. This enables it to employ code optimizations that are available in CPUs. Hence, it is possible to implement SpMM and SDDMM kernels with code optimizations such as loop reordering with a one-to-one mapping. Similarly, loop strip-ming and tiling can be mapped. However, similar to SPADE, where we accounted for the "barrier" optimization in the mapping process, we need to consider the scratchpad reuse here.
>
> Flexagon is another reconfigurable accelerator for sparse computations. Flexagon’s configurations can be mapped to loop reordering as three different data flow models use three loop traversal orders. Further, we can consider the usage of different formats of different matrices in Flexagon as a heterogeneous feature for the hardware platform. Although Flexagon has mentioned that they are using “tiling”, they have not provided a detailed implementation of it (source code is not available) or how tile size selection is done. However, considering standard tiling implementation, we are confident that we can represent Flexagon’s tiling mechanism using loop-stripping and loop reordering.
>
> Unfortunately, Intel PIUMA is proprietary, and Flexagon’s source code was not available, so we were unable to test our hypothesis on these accelerators. SPADE was the only accelerator we were able to gain access to, and we used it for our evaluations to demonstrate that heterogeneous transfer can be effectively achieved in the domain of sparse accelerators using few-shot learning.
>
> > ***Question 02*** \
> >  DASH benefits from a deeper DNN, mapping the code transformation configurations into an embedding space and transfer learning. These all seem to be standard ML techniques.
>
> **Response:** As we have explained in our response to “Weakness 02”, our innovations lie in using concepts from deep neural networks to develop a cost model architecture capable of leveraging homogeneity (through approximate mappings of code optimizations) while mitigating heterogeneity (using latent representations via an autoencoder) of program configurations across different programming systems. WACO [4], a prior work in the domain that we use as our base model, addressed the specific challenges of building cost models for sparse matrix computations. It considered the coupled behavior of the sparsity pattern, the format, and the schedule. Specifically, it introduced the use of sparse convolutional networks to capture meaningful features from sparsity patterns. In our work, we build upon WACO’s cost model (hardware-specific) to develop a cost model architecture that is amenable to heterogeneous transfer learning across different hardware platforms.
>
> > ***Question 03*** \
> >  The default optimization used by TACO and SparseTIR as the baseline
>
> **Response:** For SparseTIR's default optimizations, we used loop reordering, loop strip-mining (splitting), loop binding, and loop unrolling parameters from the example provided in the repository as our baseline. In the case of TACO, our goal was not to evaluate its performance on the CPU but rather to facilitate knowledge transfer from the CPU to other platforms. To achieve this, we focused on generating data samples with diverse parameterizations for loop reordering, loop strip-mining, and format reordering, similar to how WACO utilized TACO.
>
> > ***Question 04*** \
> >  Did you use parallelization in the evaluation?
>
> **Response:** Yes. For a single kernel execution in the CPU, we used parallelize() code optimization in TACO.

---

> > ### Author Response · Authors · 2024-11-20
> > **Response to Reviewer R7XU (part 3)**
> >
> > [1] Aananthakrishnan, Sriram, et al. "The Intel programmable and integrated unified memory architecture graph analytics processor." IEEE Micro 43.5 (2023): 78-87.
> >
> > [2] Gerogiannis, Gerasimos, et al. "HotTiles: Accelerating SpMM with Heterogeneous Accelerator Architectures." 2024 IEEE International Symposium on High-Performance Computer Architecture (HPCA). IEEE, 2024.
> >
> > [3] Muñoz-Martínez, Francisco, et al. "Flexagon: A multi-dataflow sparse-sparse matrix multiplication accelerator for efficient dnn processing." Proceedings of the 28th ACM International Conference on Architectural Support for Programming Languages and Operating Systems, Volume 3. 2023.
> >
> > [4] Won, Jaeyeon, et al. "WACO: learning workload-aware co-optimization of the format and schedule of a sparse tensor program." Proceedings of the 28th ACM International Conference on Architectural Support for Programming Languages and Operating Systems, Volume 2. 2023.

---

> ### Author Response · Authors · 2024-11-26
> **Response to Reviewer R7XU (part 4)**
>
> We sincerely thank the reviewer for taking the time to carefully consider our responses and for sharing your valuable insights. We appreciate the opportunity to further clarify our contributions and apologize for any prior lack of clarity. Below, we have addressed the concerns raised by the reviewer.  Please feel free to let us know if any aspects require further clarification. We remain open to any additional discussions you may have.
>
> > ***Concern 01*** \
> > the proposed approach relies heavy user intervention
>
> **Response:** In Figures 2 and 4, we demonstrate the necessity of user intervention (approximate mapping of code optimizations) to achieve effective knowledge transfer across heterogeneous hardware platforms. DASH achieves superior speedups compared to techniques without user intervention like feature augmentation. These approximations enabled us to overcome the limitations of prior work in the domain, which achieved effective knowledge transfer only between hardware platforms of the same architecture [1,2,3].
>
> Further, as we have explained the intuition behind these approximations when detailing the portability of DASH in our response to “Question 01”, our approximations mainly focus on loop transformations, a widely used and well-understood class of code optimizations. Therefore, making these approximations is not overly complex. We consider automating these approximations an exciting direction for future work and appreciate the reviewer for emphasizing its importance.
>
> > ***Concern 02*** \
> > the performance of the proposed appears to be largely depend on the capability of a deep neural network
>
> **Response:** Our contributions lie in **using** concepts from deep neural networks to develop a cost model architecture capable of leveraging homogeneity (through embeddings of approximate mappings of code optimizations) while mitigating heterogeneity (using latent representations via an autoencoder) of program configurations across different programming systems. In Figures 4 and 7, we provide empirical evidence that inclusion of these innovations (DASH Top-1) outperform conventional techniques (WACO+FA, WACO+FM).
>
> Moreover, in our work, we explicitly demonstrate that existing model architectures are insufficient in their current form (Figure 2) and our innovations are necessary to address the unique challenges of heterogeneous hardware platforms. We will further provide justifications for the selection of various components in the cost model design in our revised submission.
>
> Once again, we sincerely thank the reviewer for their thoughtful feedback and valuable suggestions. Your insights have greatly contributed to refining our work.
>
> [1] Sasaki, Yuta, et al. "A Cost Model for Compilers Based on Transfer Learning." 2022 IEEE International Parallel and Distributed Processing Symposium Workshops (IPDPSW). IEEE, 2022.
>
> [2] Won, Jaeyeon, et al. "WACO: learning workload-aware co-optimization of the format and schedule of a sparse tensor program." Proceedings of the 28th ACM International Conference on Architectural Support for Programming Languages and Operating Systems, Volume 2. 2023.
>
> [3] Zheng, Lianmin, et al. "Tenset: A large-scale program performance dataset for learned tensor compilers." Thirty-fifth Conference on Neural Information Processing Systems Datasets and Benchmarks Track (Round 1). 2021.

---

### Author Response · Authors · 2024-11-20
**General Comment**

We would like to sincerely thank all the reviewers for their time and effort in carefully reviewing our contributions and providing valuable feedback. Your insights are greatly appreciated, and we have worked diligently to address concerns and suggestions raised by each reviewer separately to the best of our ability within the given timeframe. Also, we would like to clarify and address some common concerns raised, as outlined below.

**Contribution:** In this work, our primary contribution is the integration of machine learning-based optimization techniques ( learned cost models) to enable effective heterogeneous knowledge transfer for emerging hardware platforms (e.g., sparse accelerators). These models assist in making critical design choices (e.g. cache sizes, memory hierarchies, and compute units) during the early stages of development of emerging hardware.

**Impact:** Our work will empower hardware architects to fully harness the potential of emerging hardware platforms by enabling them to perform comprehensive design space exploration (DSE) as the automatic selection of the best software program configuration for a given input in a hardware setup removes a dimension of complexity from the DSE stage reducing expensive simulation costs.

In addition, we will be including the following experimental results in our revised submission (to be submitted by Nov 27th).

- End-to-end results for GPU execution for a GNN with DASH.
- An ablation study to justify the choice of autoencoders in DASH.
- An ablation study to justify the choice of MLP as a predictor in DASH.

---

### Comment · Area_Chair_3qut · 2024-11-23
**Reminder: Please Review Author Responses**

Dear Reviewers,

As the discussion period is coming to a close, please take a moment to review the authors’ responses if you haven’t done so already. Even if you decide not to update your evaluation, kindly confirm that you have reviewed the responses and that they do not change your assessment.

Thank you for your time and effort!

Best regards, AC

---

### Author Response · Authors · 2024-11-24
**General Comment**

Dear Reviewers,

We are sincerely thankful to all the reviewers for your feedback and constructive comments. We have carefully addressed the points raised in our responses and currently working on revisions based on your suggestions. If any aspects remain unclear or require further clarification, we would be happy to provide additional explanations.

Best Regards,
Authors

---

### Author Response · Authors · 2024-11-28
**General Comment and Changes in Our Revision**

We sincerely thank all reviewers for your insightful comments, constructive suggestions, and actively engaging in discussions. We have uploaded the revised paper, carefully addressing the reviewers' concerns and incorporating additional experiments to the best of our ability within the provided time frame.

We appreciate that **all** reviewers have recognized the **importance of the problem** (Reviewers R7XU, rzMd, bgES and 5VNi) that we are solving and the ability of DASH to **deliver better speedups** (Reviewers R7XU, rzMd, bgES and 5VNi) with limited data compared to existing techniques. We are also encouraged by the reviewers’ acknowledgment of the **novelty** of this work  (Reviewers rzMd, bgES and 5VNi), **thoroughness of** our **experiments** and analysis (Reviewer rzMd), and **generalization capabilities** of DASH to other hardware (Reviewer rzMd and bgES). Additionally, we appreciate the positive feedback on the **quality of writing** (Reviewer rzMd and 5VNi).

Below, we summarize the newly added experiments included in the revised paper:

1. An ablation study to explore different predictors in the cost model design (Reviewer rzMd and R7XU)
2. An ablation study to explore different methods to tackle the heterogeneity of the program configurations (Reviewers rzMd, bgES and R7XU)
3. Comparison of DASH’s performance in GPUs with cusparseSpMM (Reviewer bgES)
4. End-to-end performance of DASH in GPUs for graph neural networks (Reviewer R7XU)

In addition to the above experiments, we have ensured that the revised paper has been updated to provide greater clarity in addressing all feedback received. Specifically, we have improved the introduction section to address Reviewer 5VNi’s comments and suggestions emphasizing the importance of automatic program configuration selection during the early stages of accelerator development.

We deeply appreciate thoughtful feedback of all reviewers, which has helped us strengthen the paper. Please let us know if you have any further questions or comments. Thank you!

---

### Meta-Review · Area_Chair_3qut · 2024-12-07

**Metareview:**

This work explores the use of machine learning to develop a cost model for sparse matrix computations on hardware systems, with the goal of accelerating hardware computation optimization. The approach relies on domain-specific feature engineering, leveraging the homogeneity of inputs and heterogeneity of hardware platforms to enhance the model's generalization. The methodology involves generating low-cost data samples on widely available general-purpose hardware, followed by few-shot fine-tuning to adapt efficiently to emerging hardware platforms.

Strengths:
1. The work addresses an interesting application area with practical relevance.
2. It effectively combines domain knowledge with deep learning techniques.
3. The empirical results demonstrate promising performance.

Weaknesses:

1. The evaluation is limited to a single emerging hardware platform, which restricts the generality and impact of the results.
2. The proposed approach for improving transferability might not generalize to broader settings, limiting its applicability to a wider audience.

Overall, this work presents both strengths and weaknesses, making it a borderline submission. While it adds value in its specific application domain, the narrow evaluation scope and highly domain-specific methodological insights limit its relevance to the broader ICLR community, which emphasizes generalizable and widely applicable contributions. Considering ICLR's high standards, I lean toward rejecting this submission. I strongly recommend the authors expand the evaluation scenarios to demonstrate the broader applicability of their approach. Moreover, the specific focus on hardware acceleration and sparse matrix computations may align better with venues specializing in hardware-oriented research and applications.

**Additional Comments On Reviewer Discussion:**

After the rebuttal, three out of four reviewers weakly support acceptance, while one leans towards rejection. Two of the positive reviewers expressed concerns about the narrow evaluation on a specific hardware platform, a sentiment I share. Although the authors provided some explanations during the rebuttal, these responses lack concrete evidence to substantiate their claims.

The negative reviewer mainly concerned about the heavy reliance on human intervention, which I interpret as a strong dependence on domain expert knowledge. I agree with the authors' rebuttal that such expert knowledge is essential for achieving positive transfer learning. However, the authors could improve their explanation by highlighting the general principles or insights underlying this process. Doing so might help the broader ICLR community better appreciate the contribution and its potential implications.

---

### Decision · Program_Chairs · 2025-01-22

Reject